# The Greater Caucasus Glacier Inventory (Russia/Georgia/Azerbaijan)

Levan G. Tielidze[1,3], Roger D. Wheate[2]

[1]Department of Geomorphology, Vakhushti Bagrationi Institute of Geography, Ivane Javakhishvili Tbilisi State University, 6 Tamarashvili st., Tbilisi, Georgia, 0177

[2]Natural Resources and Environmental Studies, University of Northern British Columbia, 3333 University Way, Prince George, BC, Canada, V2N 4Z9

[3]Department of Earth Sciences, Georgian National Academy of Sciences, 52 Rustaveli Ave., Tbilisi, Georgia, 0108

*Correspondence to:* Levan G. Tielidze (levan.tielidze@tsu.ge)

## Abstract

There have been numerous studies of glaciers in the Greater Caucasus, but none that have generated a modern glacier database across the whole mountain range. Here, we present an updated and expanded glacier inventory at three time periods (1960, 1986, 2014) covering the entire Greater Caucasus. Large scale topographic maps and satellite imagery (Corona, Landsat 5, Landsat 8 and ASTER) were used to conduct a remote sensing survey of glacier change and the 30 m resolution ASTER GDEM (17/11/2011) to determine the aspect, slope and height distribution of glaciers. Glacier margins were mapped manually and reveal that in 1960, the mountains contained 2349 glaciers with a total glacier surface area of $1674.9 \pm 70.4$ km$^2$. By 1986, glacier surface area had decreased to $1482.1 \pm 64.4$ km$^2$ (2209 glaciers), and by 2014 to $1193.2 \pm 54.0$ km$^2$ (2020 glaciers). This represents a $28.8 \pm 4.4\%$ ($481 \pm 21.2$ km$^2$) or 0.53% yr$^{-1}$ reduction in total glacier surface area between 1960 and 2014 and an increase in the rate of area loss since 1986 (0.69% yr$^{-1}$), compared to 1960-1986 (0.44% yr$^{-1}$). Glacier mean size decreased from 0.70 km$^2$ in 1960 to 0.66 km$^2$ in 1986 and to 0.57 km$^2$ in 2014. This new glacier inventory has been submitted to GLIMS, and can be used as a basis dataset for future studies.

## 1 Introduction

Glacier inventories provide the basis for further studies on mass balance and volume change, which are relevant for local to regional-scale hydrological studies (Huss, 2012; Fischer et al., 2015), and to global calculation of sea level change (Gardner et al., 2013; Radic and Hock, 2014). In addition, glacier inventories are invaluable data sets for revealing the characteristics of glacier distribution and for upscaling measurements from selected locations to entire mountain ranges (Nagai et al., 2016).

In a high mountain system such as the Greater Caucasus, glaciers are an important source of water for agricultural production, and runoff supplies several hydroelectric power stations. Most rivers originate in the mountains and the melting of glaciers/snow are important component inputs in terms of water supply and for recreational opportunities (Tielidze, 2017). However, glacier hazards are relatively common in this region leading to major loss of life. On September 20 2002, for example, Kolka Glacier (North Ossetia) initiated a catastrophic ice-debris flow killing over 100 people (Evans et al., 2009), and on May 17 2014, Devdoraki Glacier (Georgia) caused a rock-ice avalanche and glacial mudflow killing nine people (Tielidze, 2017). The Greater Caucasus glaciers also have economic importance as a major tourist attraction e.g. Svaneti, Racha and Kazbegi regions in Georgia, with thousands of visitors each year (Georgian National Tourism Administration, 2017).

The GLIMS database (9.02.2017) for the Greater Caucasus identified in excess of 1295 glaciers with a combined area of 1111.8 km$^2$ but with some inconsistent registration. The RGI5 database identifies in excess of 1638 glaciers with a combined area of 1276.9 km$^2$ and incorporates nominal glaciers as circles in the eastern and western Caucasus sections (i.e. no outline extents) from the WGI-XF (Cogley, 2009); these are omitted from the GLIMS database. As nobody volunteered to write a section about the glaciers in the Caucasus for the GLIMS book (Kargel et al., 2014), the region is missing in this compilation.

Thus, the objectives of this paper are to construct an updated glacier inventory for the Greater Caucasus region based on manual delineation of glaciers from multi-temporal satellite images, and especially to fill the gap in the eastern Greater Caucasus.

## 2 Study Area

The Caucasus mountains consist of two separate mountain systems: the Greater Caucasus (the higher and more extensive part) extends for ~1300 km from northwest to southeast between the Black and Caspian seas, while the Lesser Caucasus, approximately 100 km to the south is characterized by relatively lower elevations. The Greater and Lesser Caucasus are connected by the Likhi range, which represents the watershed between the Black and Caspian seas.

The Greater Caucasus can be divided into western, central and eastern sections based on morphology divided by the mountains Elbrus (5642 m) and Kazbegi (5047 m) (Maruashvili, 1981) (Fig. 1). At the same time, the terms Northern and Southern Caucasus are frequently used to refer to the corresponding macroslopes of the Greater Caucasus range (Solomina et al., 2016).

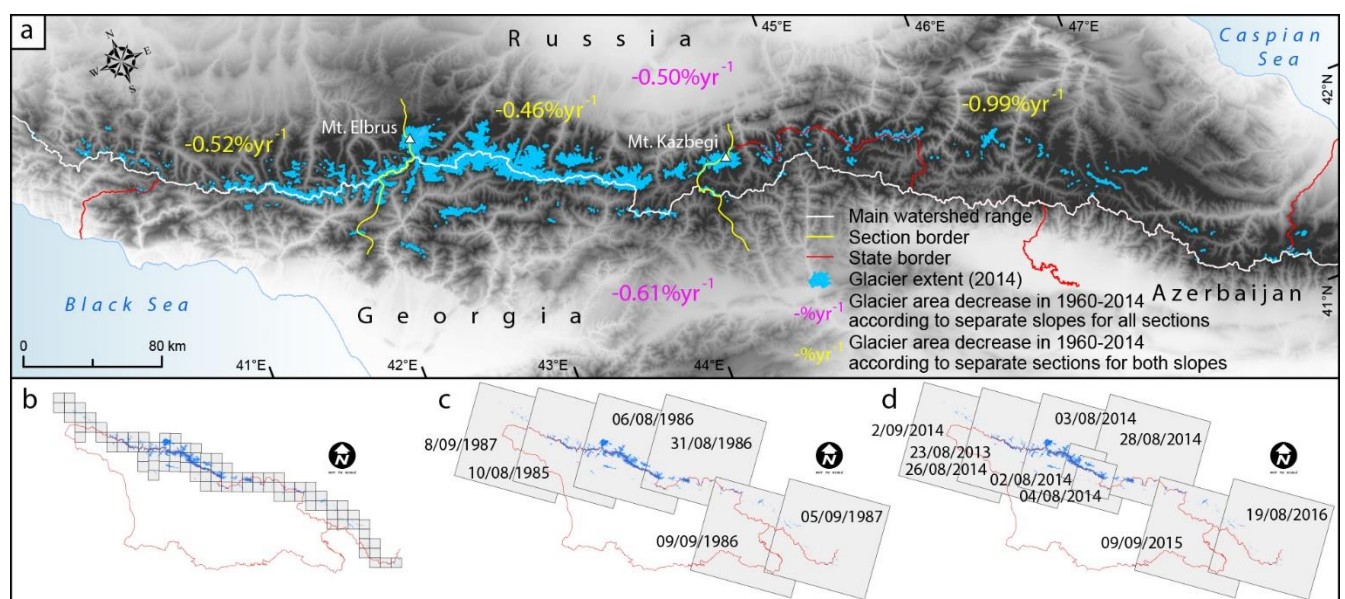

**Figure 1.** a - Distribution of the Greater Caucasus glaciers, b - 1960s 1:50 000 scale map sheets (88) are based on aerial photographs 1950-1960. c - Six Landsat 5 TM satellite scenes 1985-1987. d - Seven Landsat 8 OLI satellite scenes from 2013 to 2016 and two (smaller) ASTER satellite scenes from 2014.

Glacier retreat started from the Little Ice Age (LIA) maximum positions in the northern Caucasus from the late 1840s, with minor readvances in the 1860s-1880s and readvances or steady states in the 20th century (1910s, 1920s and 1970s-1980s) (Solomina, et al., 2016).

In the Caucasus, supra-glacial debris cover has a smaller extent than in many glacierized regions, especially Asia (Stokes et al., 2007; Shahgedanova et al., 2014). Direct field monitoring reveals evident

debris expansion for some glaciers (e.g. Djankuat) from 2% to 13% between 1968-2010 (Popovnin et al., 2015). Glacier retreat appears to be associated with expansion of supraglacial debris cover and ice-contact/proglacial lakes, which may increase the likelihood of glacier-related hazards and debris flows (Stokes et al., 2007). Debris cover is more common in the north than in the south (Lambrecht et al., 2011, Tielidze et al., 2017).

## 2.1 Previous studies

The study of glaciers in the Caucasus began in the first quarter of the 18th century, in the works of Georgian scientist Vakhushti Bagrationi (Tielidze, 2016); subsequently there were many early expeditions and glacier photographs covering the time period 1875-1906 (Solomina et al., 2016). Studies focused on glacier mapping, began when Podozerskiy (1911) published the first inventory of the Caucasus glaciers, based on large scale military topographical maps (1:42 000) from 1881-1910, identifying 1329 glaciers, with total area 1967.4 $km^2$ in the Greater Caucasus (Kotlyakov et al., 2015). Detailed analysis of these early data showed some defects in the depicted shape of the glaciers and in particular those in inaccessible valley glaciers (Tielidze, 2016). Reinhardt (1916, 1936) noted Podozerskiy's errors in compiling a new catalog for some glacial basins of the Greater Caucasus region (Tielidze, 2017).

The next inventory of the Caucasus glaciers (Catalog of Glaciers of the USSR, The Caucasus, 1967-1978), assessed glacier parameters from ~1950-1960 aerial photographs. This includes some errors as temporary snowfields were misinterpreted as glaciers (Gobejishvili, 1995; Tielidze, 2016) and the catalog datasets contained glacier parameters but not outlines. As the USSR catalog and 1960s large-scale (1:50 000) topographic maps were based on the same aerial photographs, we have used both datasets in this article for a more comprehensive comparison.

Gobejishvili (1995) documented further statistical information about the glaciers of Georgia based on the same 1960s topographic maps, reporting there were 786 glaciers with a total area of 563.7 $km^2$ in the Georgian Caucasus. The current investigation revealed that he missed some small glaciers, particularly in the Bzipi, Kodori, Rioni, Enguri and Tergi river basins.

Khromova et al. (2009, 2014) used manually digitized results to estimate changes of more than 1200 glaciers in the Caucasus between three periods - 1911-1957-2000. They found that glacier area decreased from 1911-1957 by 24.7% (0.52% $yr^{-1}$) and from 1957-2000 by 17.7% (0.41% $yr^{-1}$). Elbrus glaciers lost 14.8% (0.31% $yr^{-1}$) and 6.3% (0.14% $yr^{-1}$) respectively for the two time periods. However, there was a difference between north and south slopes of the Caucasus. Glacier area change on the north slope was 30% (0.63% $yr^{-1}$) for the first part of the 20th century and 17.9% for the second part (0.41% $yr^{-1}$). In contrast, the south slope lost 12% (0.25% $yr^{-1}$) and 28% respectively (0.65% $yr^{-1}$) indicating a contrasting slow-down/increase of the loss rate on the northern/southern slopes.

Lur'e and Panov (2014) examined northern Caucasus glacier variation for 1895-2011, finding glacier area decreased by 849 $km^2$ (52.6%). During this period, the average rate of glacier area reduction was 0.45%$yr^{-1}$, varying from 0.52%$yr^{-1}$ in 1895-1970 to 0.32%$yr^{-1}$ in 1971-2011. The most significant decrease was registered in the basins of Dagestan rivers (eastern Caucasus section); however, they didn't describe the data sources for their glacier mapping.

The most recent glacier inventory, based on old topographic maps (1911/1960) and modern satellite imagery (Landsat/ASTER, 2014) was published by Tielidze (2016), but compiled only for Georgian Caucasus glaciers, which reduced from 613.6±9.8 $km^2$ to 355.8±8.3 $km^2$ (0.37% $yr^{-1}$) between 1911-2014, while glacier numbers increased from 515 to 637. The current investigation has revealed, that some small glaciers were omitted as Tielidze used Gobejishvili's (1995) glacier database.

Other recent published works about the Caucasus, have mainly examined changes in glacier area and length for individual river basins or separate sections. Stokes et al. (2006; 2007) determined that 94% of 113 selected glaciers in the central Caucasus retreated between 1985 and 2000; the largest glaciers (>10 km$^2$) had retreated at twice the rate (~12 m yr$^{-1}$) as the smallest (<1 km$^2$) glaciers (~6 m yr$^{-1}$). Shahgedanova et al. (2014) calculated 4.7±2.1% or 0.20% yr$^{-1}$ glacier area loss from 407.3±5.4 km$^2$ to 388.1±5.2 km$^2$ in the central and western Greater Caucasus, between 1987-2010.

In this article, we present the percentage and quantitative changes in the number and area of glaciers for the whole Greater Caucasus in the years 1960, 1986 and 2014, including analyses of various glacier attributes (aspect, slope) and location.

## 3 Methods

### 3.1 Data sources

We utilise increasingly accessible global satellite imagery (Wulder et al., 2012; 2016; Pope et al., 2014) to investigate glacier area and number change in the Greater Caucasus between 1960-1986-2014. Changes in glacier extent in the Greater Caucasus between 1986 and 2014 were determined through analysis of images from Landsat 5 Thematic Mapper (TM), Landsat 8 Operational Land Imager (OLI), and the Advanced Spaceborne Thermal Emission and Reflection Radiometer (ASTER) (Table 1). Georeferenced images were downloaded using the Earthexplorer (http://earthexplorer.usgs.gov/) and Reverb/ECHO tools (http://reverb.echo.nasa.gov/).

We used the Landsat 8 panchromatic band, along with a color-composite scene for each acquisition date, combining shortwave-infrared, near-infrared, and red for Landsat, and near-infrared, red and green for ASTER images. These false-colour composite images can accurately show many glacier termini where meltwater streams display as bright blue and contrast with the snout which casts an obvious shadow (Stokes at al., 2006). This contrast remains apparent even with significant glacier retreat.

All images were acquired at the end of the ablation season, ranging from 2 August to 9 September, when glaciers were mostly free of seasonal snow under cloud-free conditions, but with some glacier margins obscured by shadows from rock faces and glacier cirque walls. In total, six Landsat 5 (TM) scenes were used for 1985/86/87, with seven Landsat 8 (OLI) for 2013/14/15/16 and two ASTER scenes used for 2014 (Fig. 1c, d; Table 1). The latter were used primarily to complete coverage from isolated cloud cover in the Landsat scenes.

Large-scale topographic maps (88 sheets, 1:50 000 scale) with a contour interval of 20 m from several hundred aerial photographs taken between 1950-1960 were used to evaluate glacier outlines (Fig. 1b). Corona images, dating from 1964 were obtained from the Earthexplorer website (http://earthexplorer.usgs.gov/) and two were georectified for comparison with map extents. As the maps were only available in printed form, we scanned at 300 dpi with 5 m ground resolution and with the Corona imagery co-registered using the 3 August 2014 Landsat image as a master (Tielidze, 2016). Offsets between the images and the Corona/archival maps were within one pixel (15 m) based on an analysis of common features identifiable in each dataset. We reprojected Corona imagery and maps to Universal Transverse Mercator (UTM), zones 37/38-north on the WGS84 ellipsoid, to facilitate comparison with modern image datasets (ArcGIS 10.2.1). Together with Landsat imagery, these older topographic maps and Corona imagery enabled us to identify changes in the number and area of glaciers over the last half century.

The 30 m resolution ASTER Global DEM (GDEM, 17/11/2011) was used to determine the aspect, slope and height distribution of glaciers, downloaded from NASA LPDAAC Collections (http://earthexplorer.usgs.gov/).

We detected the glacier length by measurement of changes in the central flowline (Paul and Svoboda, 2009). This method uses the maximum length along the central flowline in different years, input that is required for glacier inventories.

**Table 1.** List of satellite images scenes used in this study.

| Date | Resolution | Path | Row | Type of imagery | Region/Section | Scene ID |
|------|-----------|------|-----|-----------------|----------------|----------|
| 25/08/1964 | 2 m | - | - | Corona | C Greater Caucasus | DS1115-2154DF071_d |
| 25/08/1964 | 2 m | - | - | Corona | E Greater Caucasus | DS1115-2154DA082_c |
| 10/08/1985 | 30 m | 172 | 030 | Landsat 5 | W and C Greater Caucasus | LT51720301985222XXX04 |
| 06/08/1986 | 30 m | 171 | 030 | Landsat 5 | C Greater Caucasus | LT51710301986218XXX02 |
| 31/08/1986 | 30 m | 170 | 030 | Landsat 5 | C and E Greater Caucasus | LT51700301986243XXX03 |
| 09/09/1986 | 30 m | 169 | 031 | Landsat 5 | E Greater Caucasus | LT51690311986252XXX03 |
| 05/09/1987 | 30 m | 168 | 031 | Landsat 5 | E Greater Caucasus | LT51680311987248XXX03 |
| 08/09/1987 | 30 m | 173 | 030 | Landsat 5 | W Greater Caucasus | LT51730301987251AAA04 |
| 23/08/2013 | 15/30 m | 172 | 030 | Landsat 8 | W and C Greater Caucasus | LC81720302013235LGN00 |
| 03/08/2014 | 15/30 m | 171 | 030 | Landsat 8 | C Greater Caucasus | LC81710302014215LGN00 |
| 26/08/2014 | 15/30 m | 172 | 030 | Landsat 8 | W and C Greater Caucasus | LC81720302014238LGN00 |
| 28/08/2014 | 15/30 m | 170 | 030 | Landsat 8 | C and E Greater Caucasus | LC81700302014240LGN00 |
| 02/09/2014 | 15/30 m | 173 | 030 | Landsat 8 | W Greater Caucasus | LC81730302014245LGN00 |
| 09/09/2015 | 15/30 m | 169 | 031 | Landsat 8 | E Greater Caucasus | LC81690312015252LGN00 |
| 19/08/2016 | 15/30 m | 168 | 031 | Landsat 8 | E Greater Caucasus | LC81680312016232LGN00 |
| 02/08/2014 | 15 m | 171 | 030 | ASTER | C Greater Caucasus | AST_L1T_00308022014081313_20150622105647_51181 |
| 04/08/2014 | 15 m | 170 | 030 | ASTER | C Greater Caucasus | AST_L1T_00308042014080102_20150622114958_116303 |

### 3.2 Glacier uncertainty and accuracy assessment

We have determined uncertainty with two independent methods (buffer and multiple digitization). We used a buffer method similar to Granshaw and Fountain (2006) and Bolch et al. (2010) and adopted by Tielidze (2016). The uncertainty term for the 1960 extents is based on a buffer incorporating the root-mean-square error (RMSEx,y) of the map rectification (15 m) and the digitizing uncertainty equal to the width of a contour line (15 m).

Uncertainty is introduced by the resolution of the satellite image in terms of what can be seen, and the contrast between the glacier and adjacent terrain (Stokes et al., 2013). For debris-free glacier ice that is not obscured by clouds, DeBeer and Sharp (2007) suggested that line placement uncertainty is unlikely to be larger than the resolution of the imagery, i.e. ±30 m for Landsat 5 TM and Landsat 8 OLI. This can be seen in figure 2, along with a ±1 or 2 pixel buffer for debris-covered ice. A buffer with a width of one RMSE was created along the glacier outlines and the uncertainty term was calculated as an average ratio between the original glacier areas and the areas with a buffer increment; for the 1986 images we used a buffer equal to the resolution of the data (30 m) and a similar buffer for the 2014 glacier extents. This generated an average uncertainty of the mapped glacier area of 4.6% for 2014, 4.4% for 1986 and 4.2% for 1960. Using the buffer method from Granshaw and Fountain (2006), these yield a total potential overall error of ±4.4%.

For 1986/2014 imagery we digitised the outlines of both debris-covered and debris-free ice and tested a number of well-known semi-automated techniques (band ratio TM3/TM5 and OLI4/OLI6, ratio thresholds range ≥ 2.0) (Paul and Kaab, 2005; Andreassen et al., 2008; Bolch et al., 2010) to extract glacier outlines and compare with manually generated outlines (Fig. 2).

Generally, for debris-free glaciers, automated delineation using the spectral ratio is more consistent and reproducible than manual delineation. These techniques are relatively useful for large sample sizes and/or large glaciers where manual delineation would be time-consuming (e.g. Central Greater

Caucasus), but their value can be limited by areas of glacier with supraglacial debris (e.g. Western and Eastern Greater Caucasus) (Paul et al., 2013).

Following Paul et al. (2013) to determine the precision of the digitizing, we manually digitized fifteen differently sized glaciers independently five times in the western, central and eastern Greater Caucasus to estimate 1986 and 2014 glacier area error. For debris covered glaciers (Fig. 2a), the Normalised Standard Deviation (NSD - based on delineations by multiple digitalization divided by the mean glacier area for all outlines) was 6.9% and the difference between the manually and automatically derived area was 13.41%. For debris-free glaciers (Fig. 2b, c) the NSD was 5.7% and difference between the manually and automatically derived area was 4.9%.

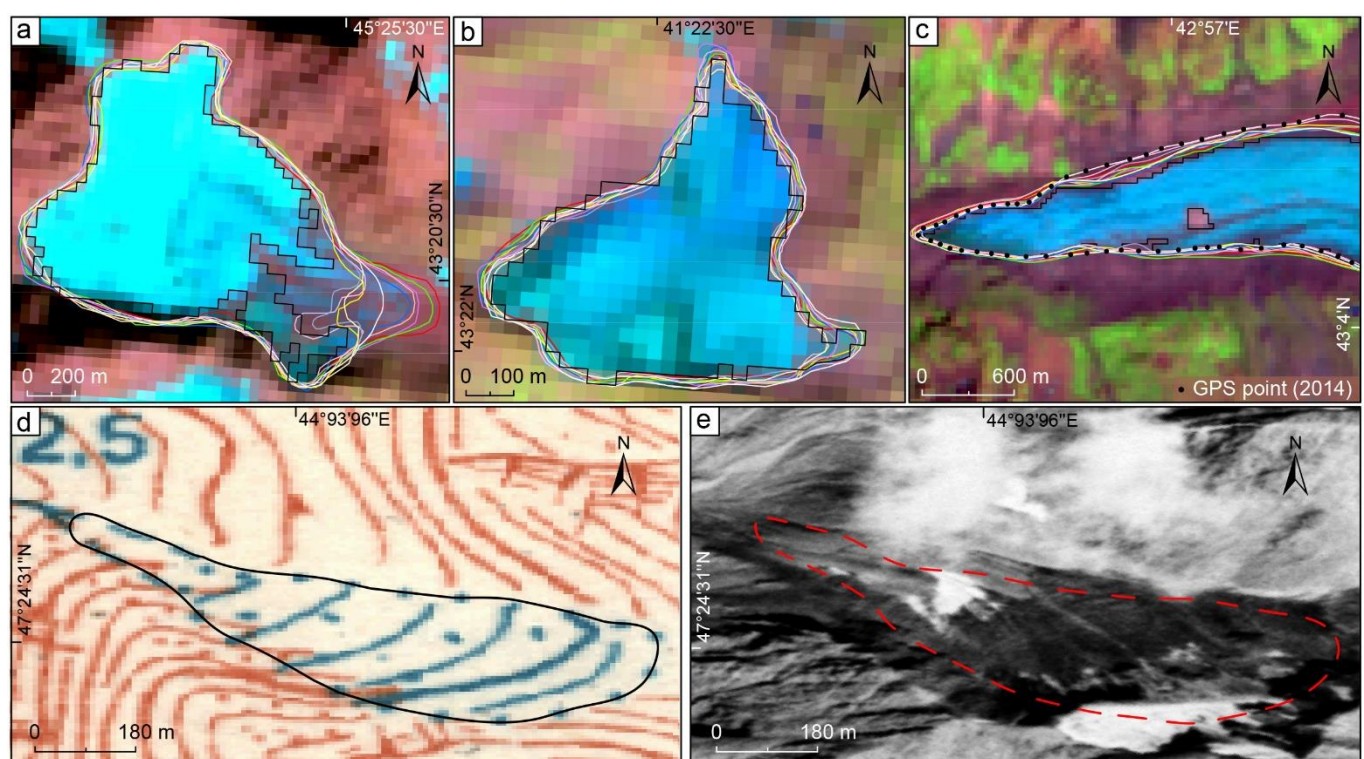

**Figure 2.** a-c Examples from the multiple digitizations of glaciers (bands 654 as RGB) using the OLI scene performed by different analysts (coloured lines). Black outlines refer to the automatically derived extents (4/6 ratio); pixel size is 30 m. d – Error in the 1960 map included mapped snow for small cirque type glacier; e – same place 1964 Corona imagery.

To estimate 1960s glacier area error we digitized multiple (3) times three different size glaciers (<2km$^2$, 2-5 km$^2$, >5 km$^2$) in the central and eastern Greater Caucasus using topographic maps and two Corona imagery (Fig. 3a-c). For 1960 topographic map glaciers, NSD was 0.4% and for Corona 5.9%. Between the maps and Corona imagery NSD was 4.8%.

Importantly, debris cover is not continuous on the snouts of many glaciers in the Greater Caucasus and most glaciers of Mt. Elbrus (Shahgedanova et al., 2014; Tielidze et al., 2017), but there are some glaciers covered by heavy debris. One of the most debris-covered we digitized in the Caucasus is Skhelda Glacier (8.28±0.65 km$^2$) (43$^o$10'N, 42$^o$38'E), where supra-glacial debris covers approximately 35% (Fig. 3d). To account for the error term due to debris cover, and following Frey et al. (2012), we increased the buffer size to two pixels (30 m) and error of mapping was calculated as ±7.9% which is the largest error in our database. In addition, for more accuracy assessment we used GPS (GARMIN 62stc) measurement data 2011-2016 for some glaciers (Fig. 2c; 3d), where GPS readings were assumed to be within one half pixel of true coordinates.

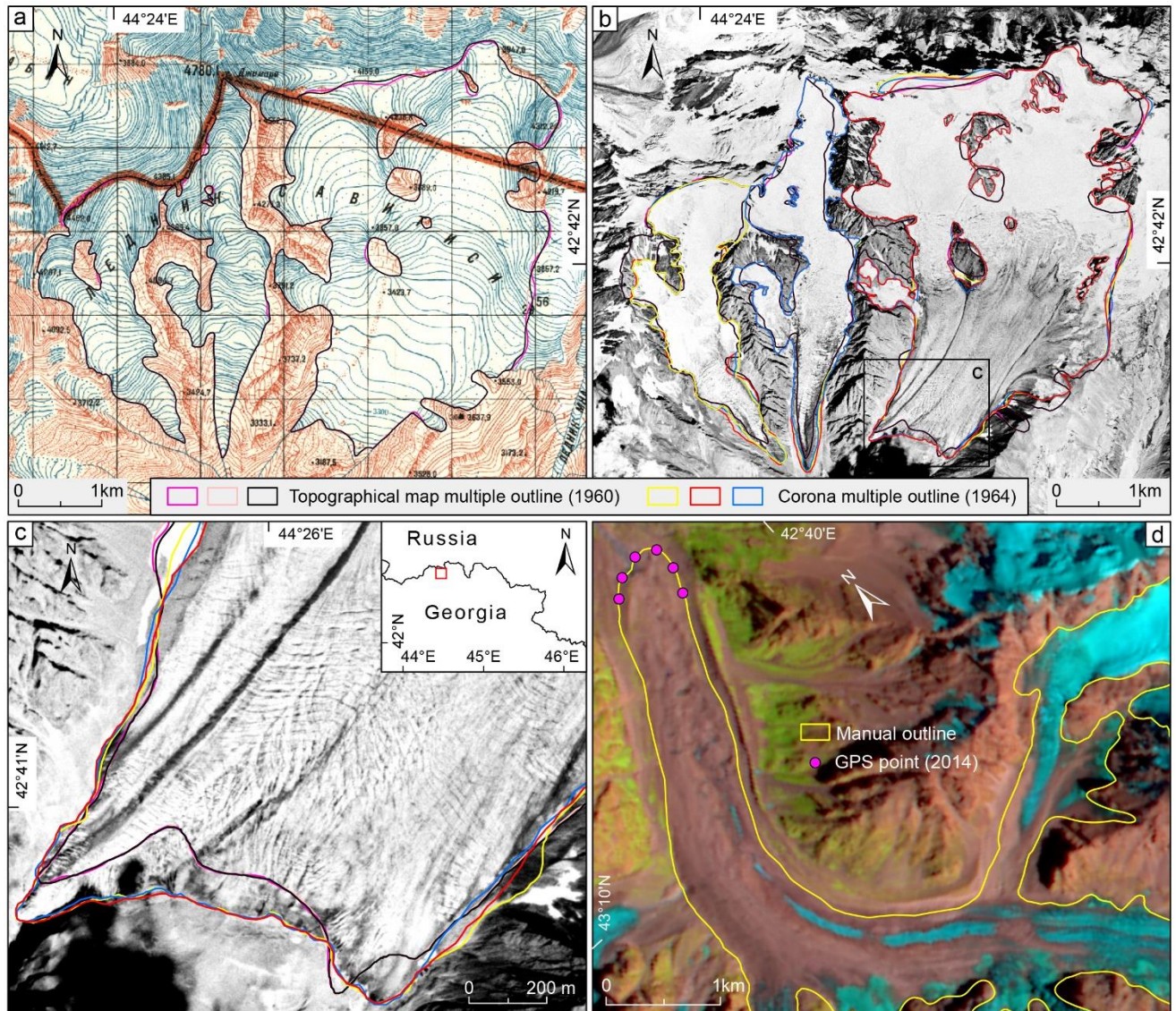

**Figure 3.** a-c Examples from the multiple digitizations of glaciers using the 1960s topographic map and 1964 Corona imagery performed by different analysts (coloured lines). d – Shkhelda Glacier, one of the heavily debris covered glaciers in the Greater Caucasus. The 23/08/2014 Landsat 8 image (Table 1) is used as background.

## 4 Results

### 4.1 Glacier changes for the entire study region

The total ice area loss between 1960 and 1986 was 192.8±8.4 km$^2$ or 11.5±4.4% (0.44% yr$^{-1}$), while the number of glaciers reduced from 2349 to 2209. Between 1986 and 2014, glacier area decreased by 288.9±12.8 km$^2$ or 19.5±4.6% (0.69% yr$^{-1}$), and glacier numbers from 2209 to 2020.

Glaciers in the northern Greater Caucasus lost 131.0±5.8 km$^2$ or 11.0±4.2% of their area (0.42% yr$^{-1}$) between 1960-1986, while the number of glaciers reduced from 1622 to 1523. Between 1986 and 2014, glacier area decreased by 189.7±8.4 km$^2$ or 18.0±4.4% (0.64% yr$^{-1}$) while the number of glaciers reduced from 1523 to 1391. Glacier mean size decreased from 0.73-0.69-0.62 km$^2$ in the northern Greater Caucasus between 1960-1986-2014.

On the southern macroslope, glacier area decreased by 61.8±3.4 km$^2$ or 12.7±5.4% (0.48% yr$^{-1}$) between 1960-1986, while the number of glaciers reduced from 727 to 686. Between 1986 and 2014,

glacier area decreased by 99.2±4.6 km$^2$ or 23.3±4.6% of their area (0.83% yr$^{-1}$) while the number of glaciers reduced from 686 to 629. Glacier mean size decreased from 0.67-0.62-0.52 km$^2$ between 1960-1986-2014.

Overall, the differences between the two macroslopes were small. The greater loss was observed on the southern slope where glaciers lost 33.0±5.0% (0.61% yr$^{-1}$) over the last half century, while the northern slope glaciers lost 27.0±4.2% (0.50% yr$^{-1}$) (Table 2; Fig. 4).

**Table 2.** The Greater Caucasus glacier number and area change according the different slopes and sections in 1960-1986-2014.

| Slope/ Section | Topographic maps 1960 | | Landsat 5, 1985/86/87 | | Landsat 8, 2013/14/15/16 and ASTER 2014 | | Decrease 1960-1986 % yr$^{-1}$ | Decrease 1986-2014 % yr$^{-1}$ | Decrease 1960-2014 % yr$^{-1}$ |
|---|---|---|---|---|---|---|---|---|---|
| | Number | Area km$^2$ | Number | Area km$^2$ | Number | Area km$^2$ | | | |
| Northern | 1622 | 1186.5±48.8 | 1523 | 1055.5±45.0 | 1391 | 865.8±38.4 | 0.42 | 0.64 | 0.50 |
| Southern | 727 | 488.4±29.0 | 686 | 426.6±19.4 | 629 | 327.4±15.2 | 0.48 | 0.83 | 0.61 |
| Western | 713 | 330.2±16.6 | 738 | 300.3±16.0 | 723 | 237.3±13.8 | 0.34 | 0.74 | 0.52 |
| Central | 1140 | 1156.0±43.2 | 1060 | 1040.5±40.0 | 1033 | 867.8±35.0 | 0.38 | 0.59 | 0.46 |
| Eastern | 496 | 188.7±10.0 | 411 | 141.6±7.8 | 264 | 88.0±5.0 | 0.96 | 1.35 | 0.98 |

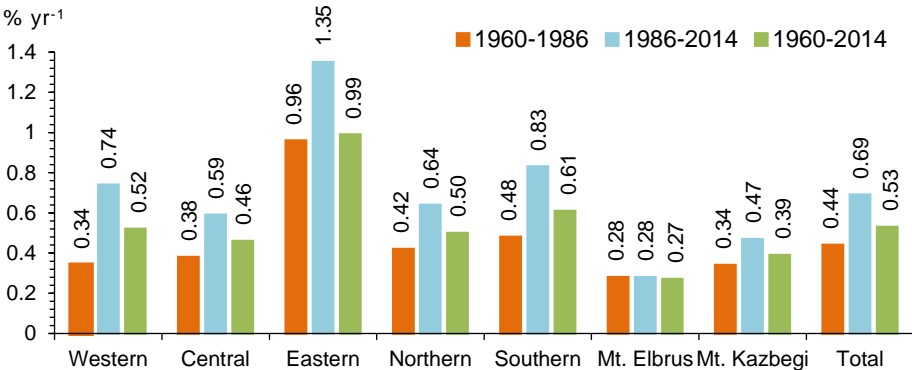

**Figure 4.** Greater Caucasus glacier area decrease by slopes, sections and mountain massifs in 1960-1986-2014.

The eastern Caucasus section (Aragvi, Tergi headwaters, Sunja right (south-east) tributaries - Sulak, Samur, Agrichai and Kusarchai) experienced the highest relative glacier area loss, where the total ice area loss between 1960 and 2014 was 53.3±4.4% (0.98% yr$^{-1}$) or 100.5±4.4 km$^2$ (Table 2; Fig. 4). Glacier area and number change by individual river basins and countries are given in the Supplement (Table 1, 2).

Glacier mean elevation for the northern macroslope changed from 3458-3477-3506 m asl between 1960-1986-2014, and minimum elevation changed 1939-1964-1997. For the southern macroslope mean elevation changed from 3246-3278-3320 m asl and minimum 1875-1908-1960 in same time. Detailed glacier parameters changes according to different slopes and sections are shown in Table 3.

### 4.2 Glacier changes on the Elbrus and Kazbegi-Dzhimara massif

Glaciers located on Mt. Elbrus lost 9.9±0.2 km$^2$ or 7.3±2.2% (0.28% yr$^{-1}$) of their combined area between 1960 and 1986 and the same amount from 1986 to 2014. Overall, the relative loss was 14.7±2.4% (0.27% yr$^{-1}$) between 1960 and 2014.

Among the large glaciers (>10 km$^2$) the Dzhikiugankez Glacier experienced a high rate of reduction, as the most extensive glacier on the Elbrus massif, the relative loss was 27.2±1.2% (0.50% yr$^{-1}$) between 1960-2014. The important relative losses for the Dzhikiugankez Glacier can be explained by the role of

post-volcanic activity, especially the influence of thermal and fluid flows in the north-eastern part of the Elbrus volcano (Masurenkov and Sobisevich, 2012; Holobâcă, 2016).

**Table 3.** Topographic parameters for glaciers 1960-1986-2014.

| Slope/ Section | Minimum elevation a.s.l. | | | Median elevation a.s.l. | | | Mean glacier size km$^2$ | | | slope° | | |
|---|---|---|---|---|---|---|---|---|---|---|---|---|
| | **1960** | **1986** | **2014** | **1960** | **1986** | **2014** | **1960** | **1986** | **2014** | **1960** | **1986** | **2014** |
| Northern | 1939 | 1964 | 1997 | 3458 | 3477 | 3506 | 0.73 | 0.69 | 0.62 | 23.22 | 22.83 | 22.21 |
| Southern | 1875 | 1908 | 1960 | 3246 | 3278 | 3320 | 0.67 | 0.62 | 0.52 | 22.78 | 22.19 | 21.16 |
| Western | 1786 | 1803 | 1891 | 3064 | 3071 | 3092 | 0.46 | 0.40 | 0.32 | 24.47 | 24.08 | 23.86 |
| Central | 1865 | 1879 | 1964 | 3453 | 3489 | 3527 | 1.01 | 0.98 | 0.84 | 22.30 | 22.02 | 21.17 |
| Eastern | 2279 | 2305 | 2332 | 3633 | 3681 | 3737 | 0.38 | 0.34 | 0.33 | 25.66 | 24.78 | 23.91 |

Unlike the Elbrus, the size of the change varied dramatically from glacier to glacier on the Kazbegi-Dzhimara massif. The total ice area loss between1960-1986 was 6.1±0.2 km$^2$ or 9.0±4.0% (0.34% yr$^{-1}$). From 1986 to 2014 glacier area decreased by 8.3±0.4 km$^2$ or 13.4±4.4% (0.47% yr$^{-1}$). Overall, the relative loss was 21.2±4.4% (0.39% yr$^{-1}$) between 1960 and 2014 (Fig. 4).

10     Among glaciers with area 2-5 km$^2$, the Devdoraki Glacier experienced a high rate of reduction between 1960-2014, with a relative loss of 38.8±2.8% (1.4% yr$^{-1}$). We do not include Kolka Glacier in a statistical analysis, as it was almost removed by its strong rock-ice avalance in 2002 (Haeberli et al., 2004; Huggel et al., 2005; Petrakov et al., 2008). Elbrus and Kazbegi-Dzhimara glaciers area and number change are given in the Supplement (Table 3-4; Fig. 1-2).

### 4.2 Glacier characteristics

The greatest area is occupied by glaciers in the size class 1.0-5.0 km$^2$ across all three time periods. (Fig. 5; in the Supplement Table 5). The largest glaciers are located in the central Greater Caucasus where valley glaciers have individual areas of 5-37 km$^2$. The total area of glaciers in the central section is more

20  than triple that in western Caucasus, which in turn is almost triple the glacier area in the eastern section, even though the eastern section is higher than the western (Table 3). For 1960 there were 22 glaciers with individual area >10 km$^2$ and for 2014 there are just 13 in the Central Greater Caucasus. There are no glaciers of >10 km$^2$ area in the eastern and western Greater Caucasus.

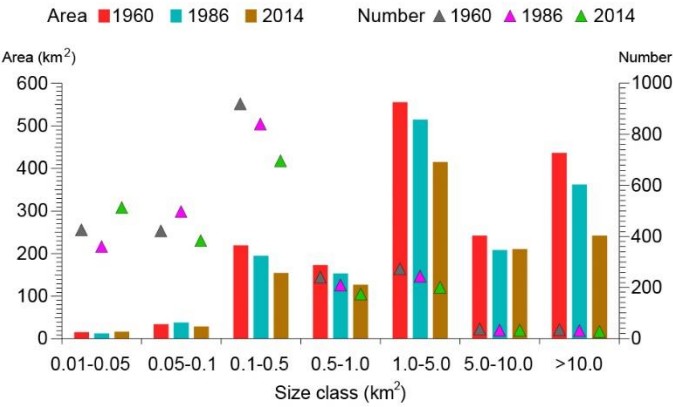

25

**Figure 5.** Cumulative glacier area and number values for seven size classes in the Greater Caucasus in 1960-1986-2014.

During the 1986-2014 period the number of smallest glaciers (0.01-0.05 km$^2$) have increased in the western and central Caucasus (1.15% yr$^+$1) at the expense of relatively larger glaciers. Glacier area

30  change by individual sections are shown in Fig. 6a-c, and in the Supplement Table 6.

Glacier area reduction varies between the individual sections of the Caucasus, with the highest increase in the number of glaciers in the smallest category (<0.05km$^2$) in the western section, resulting from the disintegration of larger glaciers. Glaciers of 0.1-10.0 km$^2$ showed the smallest decrease in the central Greater Caucasus, with the largest in the eastern Caucasus. The largest glaciers with >10 km$^2$ area all in the central Caucasus showed the greatest overall loss (0.82% yr$^{-1}$) (Fig. 6d). The Greater Caucasus cumulative glacier area and number values for seven size classes glacier in 1960-1986-2014 by individual sections and slopes are given in the Supplement (Figs. 3-12).

Most of the glacier area in the Greater Caucasus occurs between 3000 m and 4000 m a.s.l. (857.6 km$^2$), and glaciers in the northern and southern slopes are distributed mainly in the same altitudinal range. The valley glacier terminus positions are between 1900 and 3200 m., whereas cirque and hanging glaciers are at higher elevations, between 2800 and 4500 m. The distribution of glacier area with elevation is depicted in figure 7a-b.

Glaciers with north, northeast and northwest aspects are the most extensive in the Greater Caucasus, covering 286.0±12.2 km$^2$ (370 glaciers), 277.7±12.0 km$^2$ (443 glaciers) and 231.6±11.8 km$^2$ (483 glaciers) respectively, and combining for 66.7% of all glaciers (Fig. 8a, b). The south, southeast and southwest aspects cover 89.4±3.8 km$^2$ (145 glaciers), 132.7±4.8 km$^2$ (169 glaciers) and 85.0±3.4 km$^2$ (121 glaciers) respectively, and combine for 25.7% of all glaciers. The southern macroslope of the greater Caucasus is relatively shorter than the northern, providing more favourable conditions for the existence of large size glaciers in the north.

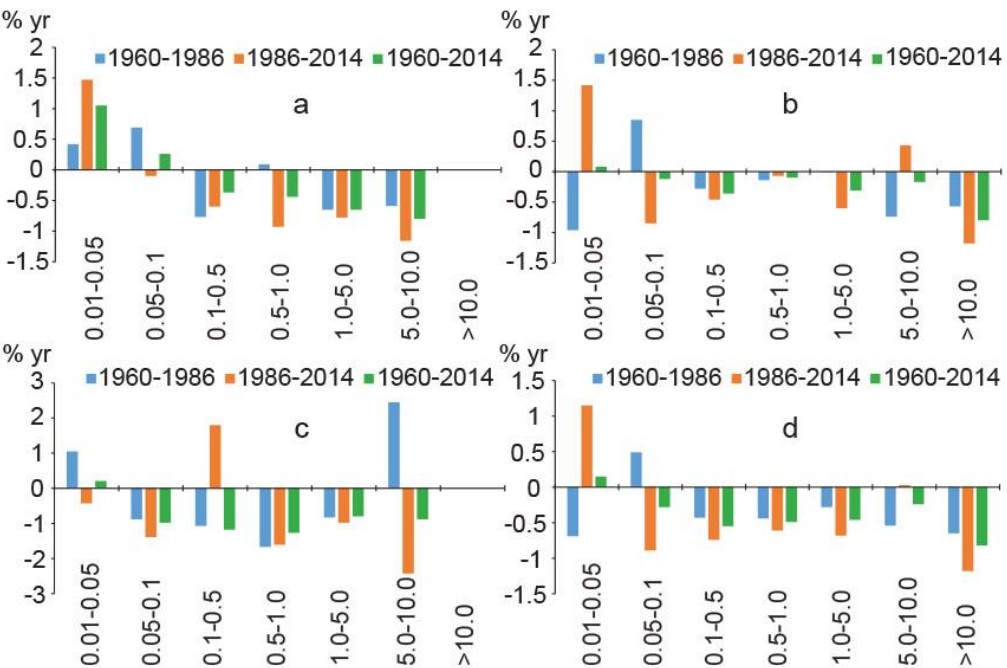

**Figure 6.** a – Area change for the seven size classes glacier in the western, b – central, c – eastern sections and d – entire Greater Caucasus in 1960-1986-2014.

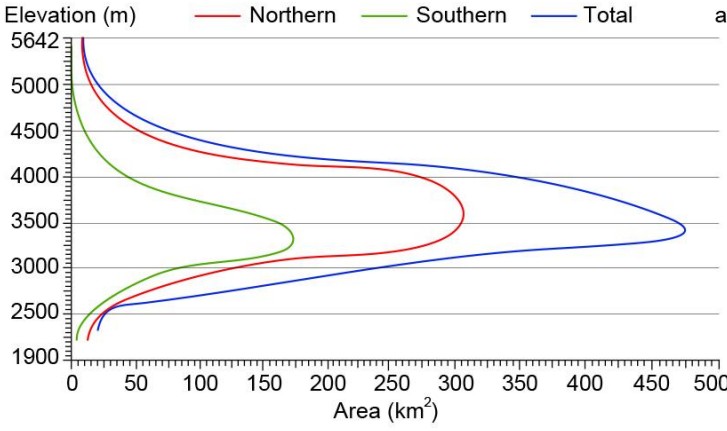

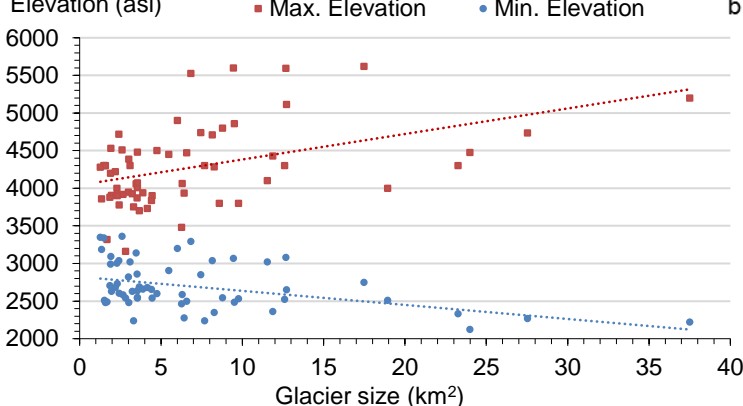

**Figure 7.** a - Distribution of glacier area with elevation for northern, southern and both slopes in the Greater Caucasus. b - Scatter plot showing glacier size vs minimum and maximum elevation

Glaciers with south aspects located on the northern slope are the most elevated in the Greater Caucasus. For the southern slope, southwest aspects are more elevated and for the entire mountain range - southeast aspects (Fig. 8c).

The slopes with 10-15º gradient are most common for both northern and southern Greater Caucasus glaciers (Fig. 9).

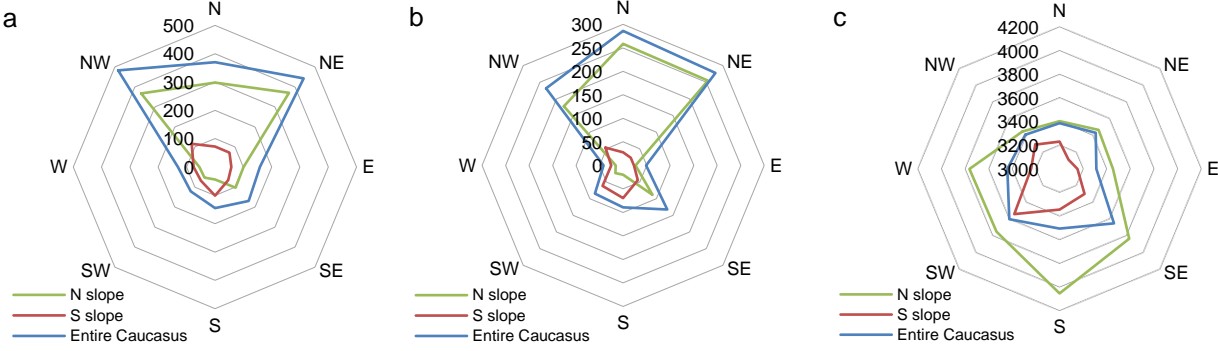

**Figure 8.** Proportion of glacier aspect by: a – number, b – area (km$^2$) and c – aspect vs mean elevation (asl).

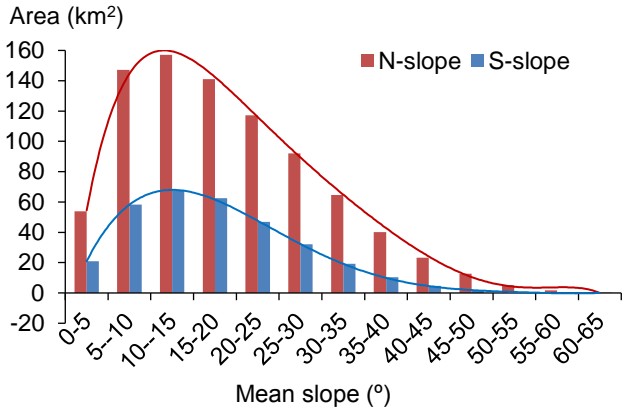

**Figure 9.** Greater Caucasus, mean slope vs glacier area for northern and southern macroslopes.

### 4.3 Glacier length change

We chose 30 glaciers in four size classes (<1 km$^2$; 1-5 km$^2$; 5-10 km$^2$; >10 km$^2$) in the western, central and eastern Caucasus macroslopes for sampling retreat rates. Across the region, retreat rates mostly increased from the 1960-1986 period to 1986-2014. The highest retreat rates of 48.8 m yr$^{-1}$ were observed on the Karaugom Glacier northern macroslope in 1986-2014, while the Shkhelda Glacier experienced double the retreat rate between 1960-1986 (36.5 m yr$^{-1}$) than in 1986-2014 (17.1 m yr$^{-1}$). The greatest total retreat was exhibited by the Lekhziri, the largest glacier on the southern macroslope, retreating 1595 m at an average rate of 29.5 m yr$^{-1}$ in 1960-2014.

Of the 30 glaciers measured, 29 retreated between 1986 and 2014. Thirteen glaciers showed less change between 1986-2014 than 1960-1986, and one glacier (Midjirgi) advanced. These results correlate well with detailed field measurement of the snout position of Chalaati Glacier (Gobejishvili, 1995; Gobejishvili et al., 2012) and are in agreement with sporadic field measurement and anecdotal evidence from other glaciers (e.g. field investigation confirms that Midjirgi Glacier advanced between 1985 and 2000). The overall advance of Mizhirgi Glacier between 1985 and 2000 was around 110±25 m (Stokes et al., 2006). Microstadial moraines in front of Chalaati Glacier confirm ~20 m glacier advance during 1990-1993 (Gobejishvili, 1995; Gobejishvili et al., 2012), but there is no clear geographical template which characterizes the advancing glaciers.

Overall, the largest glaciers (>10 km$^2$) on average retreated ~21.2 m yr$^{-1}$ between 1960-2014; glaciers with area 5-10 km$^2$ retreated ~15.6 m yr$^{-1}$; glaciers between 1-5 km$^2$ retreated ~12.0 m yr$^{-1}$ and glaciers between <1 km$^2$ - retreated ~7.8 m yr$^{-1}$.

According to this current inventory, the Bezingi Glacier represents the largest single glacier (37.47±0.94 km$^2$) in the Greater Caucasus. Characteristics of glaciers used for measuring length change are given in the Supplement (Table 7; Fig. 13, 14).

### 5 Discussion
### 5.1 Glacier inventory parameters

Considering some errors in the 1911 catalog (Tielidze, 2016), we calculate that glacier area decreased from 1967.4 km$^2$ in 1911 to 1674.9 km$^2$ in 1960 or 14.9% (0.30% yr$^{-1}$). Overall this was 39.4% (0.34% yr$^{-1}$) between 1911-2014. Thus, we consider that glacier area reduction increased between 1986-2014 in comparison with the 1960-1986 period and between 1960-2014 in comparison with the 1911-1960 period. Therefore, our results contrast with those of Lur'e and Panov (2014) - where the northern Caucasus glacier decrease was faster between 1895-1970 (0.52% yr$^{-1}$) than in 1971-2011 (0.32% yr$^{-1}$) and Khromova et al. (2009) - where the Caucasus glacier decrease was faster between 1911-1957

(0.52% yr$^{-1}$) than from 1957-2000 (0.41% yr$^{-1}$). Glacier distribution and change during the investigation period are characterized by obvious regional differences (Fig. 4). These may be related to the different topography, aspect and climatic settings of glaciers of different size, as well as climate change.

Our results also showed that eastern Greater Caucasus glaciers are shrinking faster than those in the western and central areas; the smaller size of glaciers there may be the reason for this phenomenon. The Elbrus glacier area rate of loss is lower than in the Greater Caucasus main watershed range due to the higher elevation and larger accumulation areas.

## 5.2 Glacier changes

One of the important steps in utilizing our glacier inventory data is to understand spatial patterns in glacier characteristics across the region. Our study area displays region-wide consistency in glacier characteristics, notably glacier area, elevation and topography across the five subregions based on the ~2000 glacier data (Table 3).

Comparisons between glacier size and surface area fluctuations suggest that smaller glaciers, though losing the least surface area, actually lost a greater proportion of their total area. Approximately 28.37% (0.52% yr$^{-1}$) of the glacier area that disappeared was from glaciers 0.1-1.0 km$^2$ in size between 1960-2014. Compared with similar size glaciers in the surrounding regions, e.g. the Swiss Alps, rate of glacial shrinkage was twice as high (1.13% % yr$^{-1}$ between 1973-2010) (Fischer et al., 2014). Similar trends, with small glaciers showing a propensity to shrink rapidly, have been found in numerous regions globally (Tennant et al., 2012; Stokes et al., 2013; Racoviteanu et al., 2015). This is considered a result of the greater volume-to-area and perimeter-to-area ratios of smaller glaciers – meaning they respond rapidly to a given ablation rate (Granshaw and Fountain, 2006; Tennant et al., 2012). Small glaciers are particularly sensitive to climate change. Their number will increase in the future (especially in the central Greater Caucasus) as the larger glaciers shrink and disaggregate.

Unlike the small glaciers, the largest glaciers (>10 km$^2$) are disappearing more rapidly in the Caucasus (0.82% yr$^{-1}$) than the Alps (0.60% yr$^{-1}$) (Fischer et al., 2014).

Glacier slope may also play a significant role in determining glacier area change (Table 3), i.e., the steeper the glacier, the larger the area loss observed in our study. The same tendency was observed in the Himalaya (Salerno et al., 2008; Racoviteanu et al., 2015).

We note that direct comparison of such numbers can be critical for various reasons, such as diverse sample size or size class distribution of the investigated glaciers, different subregional to local climate conditions, various length and onset of observation periods, etc..

## 5.3 Comparison to GLIMS and the RGI

The GLIMS glacier database (9.02.2017 version) contained a number of deficiencies which have been remedied after this inventory, for example these river basins did not contain any glacier outlines: Belaya, Malaya Laba, Mzimba in the western Caucasus; Khobistskali in the central Caucasus; Aragvi, Assa, Arghuni, Sharo Argun, Andiyskoye Koysu, Avarskoye Koysu, Samur, Agrichai and Kusarchai in the eastern Caucasus. These constitute more than one half of the territory for the whole Greater Caucasus where modern glaciers are present (Fig. 10a). The GLIMS outlines also involve inconsistent registration, which appears to be associated with the use of ASTER imagery (Fig. 10b) (Khromova, 2009).

The RGI 5.0 version database similarly contains errors, especially in the central Caucasus section. For example in the Samegrelo, Lechkhumi and Shoda-Kedela sub-ranges, where the RGI database contains 39 nominal glaciers (circles representing areas), with a total area of 40.2 km$^2$, we found an additional 40 glaciers with a total area of 3.5 km$^2$. In addition, almost the whole eastern Caucasus section

(except the Tergi headwaters) and some parts of the western Caucasus section (Belaya, Malaya Laba and Mzimba river basins) are represented by nominal glaciers (Fig. 10c).

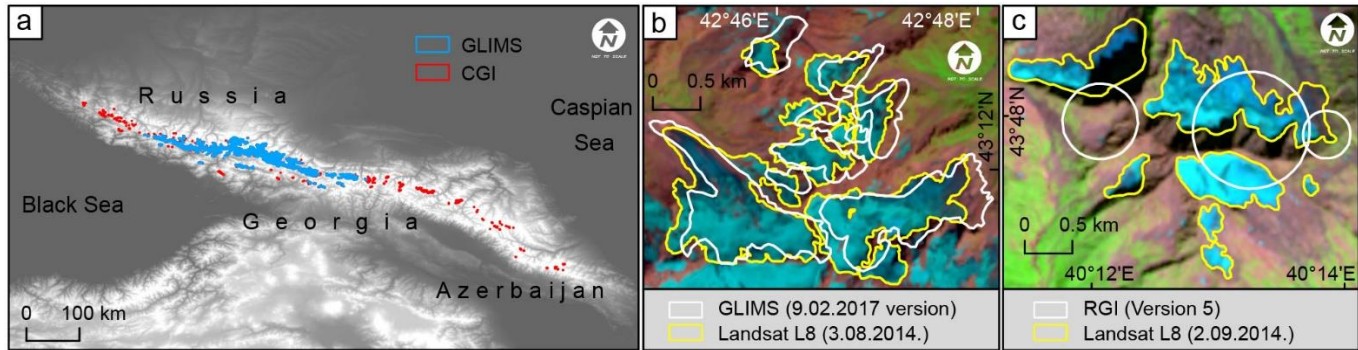

**Figure 10.** a – Comparison of the GLIMS with the new Caucasus Glacier Inventory (CGI). b – The GLIMS outlines inconsistent registration example. c – Glacier outlines from 2014, showing the RGI nominal glaciers (circles).

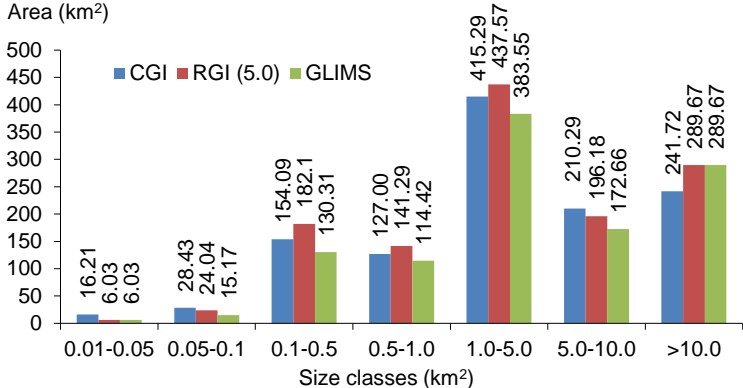

**Figure 11.** Glacier area comparison for seven classes glacier in the Greater Caucasus according the RGI (5), GLIMS and new Caucasus Glacer Inventory (CGI).

Overall, glacier area difference was 165.1 km$^2$ between the RGI (1276.9 km$^2$) and GLIMS (1111.8 km$^2$), 83.7 km$^2$ between the RGI and new Caucasus Glacier Inventory (CGI) (1193.2±54.0 km$^2$) and 81.4 km$^2$ between the GLIMS and CGI (Fig. 11).

## 6 Conclusions

We present a glacier change analysis including multi-temporal data sets covering the entire Greater Caucasus for the first time. Manual digitisation from 1960s large scale (1:50 000) topographic maps and satellite imagery from 1964 (Corona), 1986 (Landsat 5) and 2014 (Landsat 8, ASTER) were used to map glacier surface area. We expect that this inventory substantially improves existing knowledge for this region.

The main errors occur from data quality. Errors in the 1960s maps included mapped snow patches (especially for small cirque type glaciers) and uncertain glacier extents, which could be verified using available Corona 1964 satellite imagery (Fig. 2d-e). Other sources of error for aerial imagery include seasonal snow, shadows, and debris cover, which can impede glacier mapping. Using GPS field data, debris cover error can be resolved for some glaciers; while incorrect identification of seasonal snow generally affects small glaciers more than larger complexes, these do not make up a large percentage of the total area.

The main study findings can be summarised as follows:

a) The Greater Caucasus region experienced glacier area loss at an average annual rate 0.44% yr$^{-1}$ between 1960-1986 and 0.69% yr$^{-1}$ between 1986-2014. Overall, the glacier loss was 0.53% yr$^{-1}$ between 1960-2014.

b) Glacier number and area changes indicate that glaciers in the eastern Greater Caucasus have decreased (0.98% yr$^{-1}$) more than in the central (0.46% yr$^{-1}$) and western (0.52% yr$^{-1}$) sections, and southern glaciers have retreated (0.61% yr$^{-1}$) more than northern (0.50 % yr$^{-1}$) glaciers between 1960-2014. Although this rate is exceeded in other world mountain ranges (Huss and Hock, 2015), if the decrease in the surface area of glaciers in the eastern Greater Caucasus continues over the 21st century, many will disappear by 2100.

c) Glaciers of the Elbrus and Kazbegi-Dzhimara massifs lost a lower proportion of their area between 1960-2014, compared to glaciers located in the main watershed range, by 0.27% yr$^{-1}$ and 0.39% yr$^{-1}$ respectively.

The inventory presented here will further enable focus on assessing changes in glaciers, debris cover, mass balance, total volume and hydrological modeling.

## 7 Data availability

The data described in this article are available for public download at http://www.glims.org/download/

## Supplement

This Greater Caucasus Glacier Inventory includes: the number and area change in 1960-1986-2014 by individual river basins and countries (Tables 1-2). Elbrus and Kazbegi-Dzhimara massif glacier number and area change in 1960-1986-2014 (Table 3-4; Fig. 1-2); cumulative glacier area and number values for seven size classes in 1960-1986-2014 for the northern, southern, western, central and eastern Greater Caucasus (Table 5; Fig. 3-12); Area change for the seven size classes glacier in the western, central, eastern sections and entire Greater Caucasus in 1960-1986-2014 (Table 6); characteristics of glaciers used for measuring length change (Table 7).

*Acknowledgements.* We gratefully acknowledge the financial support from the Shota Rustaveli Georgian National Science Foundation "State science grants for outgoing research internship of young scientists 2016" project - The Greater Caucasus glacier inventory (IG/3/1/16).

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
