# Peer review of "The Greater Caucasus Glacier Inventory (Russia/Georgia/Azerbaijan)"

_The Cryosphere, 2017_

## Referee Comment (RC1) · F. Paul (Referee) · 21 Jun 2017

**Review of the study by Tielidze and Wheate**

**General comments**

The study by Tielidze and Wheate is presenting the results of a glacier inventory comparison for the years 1960-1986-2014 over the entire Caucasus. The authors provide a detailed overview on previous studies in the region, shortcomings of existing datasets in GLIMS/RGI, glacier sizes through time for individual river catchments and glaciers, data on cumulative glacier length change and climatic trends. Whereas I highly welcome the effort of the authors to improve on the currently available datasets for the region (in terms of consistency and spatial coverage), I am not quite sure if the study can pass scientifically. To me the material presented looks more like a data report with some limited (and partly questionable) analysis. Indeed, there is also a problem with the five more or less identical publications by the first author, each covering different river-catchments (Tielidze et al. 2015a, b, c, d) of the study region (like salami slices) or the larger part of Georgia (Tielidze 2016) and each study is presenting the datasets in the same style with limited scientific analysis. In effect, this study adds further numbers for a region previously not covered by the author but basically that's it.

With three complete glacier inventories of a larger mountain range at hand I think the authors missed the opportunity to provide a convincing scientific analysis of the data obtained. Please have a look at recent studies by Davies and Glasser (2012), Guo et al. (2015), Paul and Mölg (2014), Winsvold et al. (2014), Fischer et al. (2015), Kienholz et al. (2015), or Ojha et al. (2016) to see what is missing. Instead, I have the impression that the study is blown-up with unused details about the investigated region (e.g. the entire page 3) but important details are not given (e.g. what is the ELA on P4/L14, on which glaciers have values been measured to provide this trend analysis?). Similarly, Tables 2, 4 and 5 now cover 5 pages of text but should be presented in the Appendix. It is neither motivated why the river basins have been selected, nor why this information is important at this level of detail (e.g. input for subsequent hydrological modelling? Moreover, much space is taken in the tables for presenting uncertainties (in $km^2$ and %) for each glacier/catchment, but the area changes itself (in $km^2$ and %) are not presented (although the related results are discussed in much detail incl. the abstract). Thereby, the uncertainty values are only determined with the (statistical) buffer method. So what can we learn from all these numbers? Instead of the 350 individual values I would have liked to see one number that is related to the uncertainty of the digitizing by the analyst, e.g. based on the independent multiple digitization of a couple of glaciers. I think for completely manually digitized inventories such an assessment is mandatory.

As parts of the text have been described in previous studies or is not really required in the context of this study, I think the 'data report' presented here can be easily turned into a nice scientific paper when replacing the text that is not required with an extended analysis of the results as shown in the studies cited above.

**Specific comments**

In the following I list all comments, large and small. Some are repetitive.

P1
L18/19: 'marked acceleration': This is not obvious from the presented numbers. Please add relative change rates per year for both periods (with two decimals!)
L22: 'can be used': have/will the outlines and inventory data be submitted to GLIMS? If yes, please add so that it is clear they are accessible.

L27: The Fischer et al. (2015) study is maybe not so relevant for global sea level rise. I suggest writing 'for local to regional-scale hydrological studies (Fischer et al. 2015, Huss 2012), to global calculation of sea level change (Bliss et al. 2014, Gardner et al. 2013).

L27: 'glacial extent': Please write 'glacier extent' when referring to contemporary glaciers.

L27-31: I do not understand this sentence, please rewrite. Please note (this still goes wrong in many studies), there is no direct link between climate change and change in glacier area. Changes in area result from the combined interaction of surface lowering (which is a result of glacier mass balance) and ice thickness distribution. So for the same climatic forcing (at the same elevation) area changes can be very large when the ice is thin or small when the ice is thick. This has little to do with response times or the geometric adjustment of a glacier on the decadal time scale.

L31/2: When it is important, why has it not been done here?

L32: Tracking only the area changes over many decades might not help much. A good understanding also requires to have surface elevation changes and ice thickness distribution.

L34: Maybe the cited studies are not the best examples. Recently published glacier inventories often cover entire mountain ranges with thousands to ten-thousands of glaciers (e.g. for Alaska, Greenland, the Alps, or the Greater Himalaya).

L35: Instead of citing here Paul (2000), I suggest citing much larger and more recent studies that investigated differences between inventories (e.g. Nuimura et al. 2015 or Nagai et al. 2016).

L36: I think this sentence needs to have a connection to the sentence before (e.g. 'However, consistent methodological inventories are necessary to correctly perform change assessment and other glaciological research …'). Unfortunately, this point is not taken up again to motivate (or even justify) the inventory presented here (see next point).

P2

L7: Yes, glacier research certainly is. But why is it required to have yet another inventory of the region? Apart from several general statements, the introduction makes no attempt to motivate what is presented afterwards. When readers should also read the rest of the text, it must be clearly explained in the introduction what the research gaps are (for this region) and how this study is addressing them and presenting at the same time a never seen before analysis of the new dataset that has only become possible now. There is now a large opportunity to do so, so please do it!

L20ff: I think the text before is fine to introduce the study region. But from here until P3/L28 I suggest removing the text as the contents (descriptions of mountains and their height/location) is not used later in the text and digitally available for most of us.

P25 (Fig. 1): I am not convinced that the presentation of area changes in sub-catchments (Table 2) is really required. If the authors agree that they are not required, I would remove the catchment numbers here (but please provide them as a separate shape file with the dataset) and show instead where the glaciers depicted in Figs. 2 to 7 are located.

P3

L29ff: Some climatic background is certainly fine, but how does the information provided here help me to understand the results of the inventory, e.g. for what purpose do I need to know the minimum and maximum lapse rates or the minimum air temperature in 1983? What I want to know is mean summer or annual temperature and precipitation at the mean elevation of glaciers.

P4

L5/6: LIA history: I suggest adding here the study by Solomina et al. (2016).

L6: glaciated => glacierized

L14ff: Where does the ELA come from? ELA is related to glacier mass balance and the gradients provided here suggest that many glaciers with mass balance measurements have been used to derive it (which ones?). Note: snow lines or glacier mean elevation ≠ ELA.

L20: Please use 'retreat/advance' only when reference is made to changes in length. For mass balance one can use mass loss/gain.

L39: 786 glaciers

P5

L12/13: Please give area changes always in % as absolute values are incomparable. Moreover, a comparison of area change over periods of different length should always provide the rate (i.e. per cent per year).

L20: Fig. 2d/e: Please do some contrast enhancement here, there is not much to see.

P6

L3/4: 'retreated most dramatically': What does this mean? Is it really retreat (length change) or area change? Please provide values for such statements.

L6: Here are all the salami-slices I referred to in the general comments. Please use this publication to make some progress in the analysis. There is no need to do the same paper again and again.

L17: These references are ok but maybe somewhat out-dated? What about Wulder et al. (2012 / 2016) or Pope et al. (2014)?

L26: 'cast an obvious shadow': The problem with retreating glaciers is that their terminus does no longer 'cast an obvious shadow' and that most glaciers in the world are in retreat. So what has been done when their terminus is barely visible?

L26: manual delineation: this is correct but for clean ice automated mapping is superior (consistent and reproducible), at least in the absence of seasonal snow

L28: Not only glacier tongues need to be free of seasonal snow, the entire glacier should be.

L34: 'the most accurate method': this is certainly correct for debris-covered glaciers, but for clean ice automated mapping has the same accuracy (and outlines are consistent, not generalized and reproducible; all important assets these days).

L37: of 20 metres from 88 aerial (remove brackets)

P7

L12: I assume this should read 'To estimate uncertainty of glacier area'? Please check carefully the difference between uncertainty (of a measurement) and error (difference to a reference dataset). These are different. For example (L14), the 'digitizing error' should likely read 'digitizing uncertainty'.

L14: Please clarify how the map rectification could have an impact on the derived glacier area when considering this as an 'error term'. As far as I know, geolocation uncertainty is only an issue when directly calculated from different datasets (e.g. cumulative length changes using digital intersection). Location errors should not impact on the derived area.

L16: I think all these are not errors but uncertainties.

L18: You can see in Paul et al. (2013) how line-placing uncertainty for manual digitizing looks like. It is indeed +/-1 pixel (and worse for higher resolution imagery due to increasing generalization). A more realistic result can thus be achieved when using a +/-1 pixel buffer (and +/-2 pixels for debris-covered glaciers).

P8

L4: conservative: It is foremost just a statistical value with probably little relation to the analysts work. I would thus strongly recommend to perform a multiple (3-5 times) digitizing experiment with a couple (about 10) of differently sized glaciers (with debris and clean). This should be done for all three datasets (map, TM, ASTER) and provide a more realistic estimate of the uncertainty.

L5: Figs. 3a and b should be shown side by side. It might also e helpful to mark the boundary of Kolka Glacier in Fig. 3b as the 64% debris cover mentioned in L15 cannot be seen.

L11: This seems to be a repetition from P6/L34.

L13: 'one of the most': Why not presenting a hypsographic analysis showing where the debris is?

L14: Why not adding that Kolka has rebuild from ice avalanches from the surrounding steep rock walls after being completely removed from its bed by the 2002 avalanche? This might help to explain why it is such a special case.

L17: 'not typical': why? I think the Caucasus has quite a lot of debris-covered glaciers, in particular the larger ones.

P9

L8: As mentioned above, there is no direct link between glacier area change and climate. When climate data are analysed for the same time period you have to analyse mass balance data, only these provide a direct and undelayed response to the governing atmospheric conditions. Also for a general trend analysis the forcing has to consider glacier response times. Understanding glacier area changes for 1960 to 1986 might thus require looking at climatic conditions from 1930-1960 (for an assumed 30 year response time). But even this might only allow explaining general trends in *length* changes as area changes are also driven by the ice thickness distribution. I think it is ok to say that glaciers have lost area since the 1960s because temperatures have increased, but that's it.

L13/15 and elsewhere: please always provide relative area change rates (per year) to have comparable values. And please always give two decimals! The rounded values in L17, 19, 20, 22, and 24 are not accurate enough (see example below).

L27: 'highest glacier surface decrease': What is this? Absolute loss, relative change, change rate? Please be precise.

P10-12 (Table 2): Please rethink if this Table makes any sense. I suggest removing it completely and refer to the digital datasets for such assessments. If it should stay, move it to the Appendix and better justify why these numbers per tributary river basin are required. There is currently no further use of them. Instead of providing (statistical) uncertainties for each basin, please consider adding absolute/relative area changes. This could also be visualized graphically for all regions (using a multi-segmented bar chart) or in dependence of elevation. Please also note that numbers should be aligned right rather than left.

P12 (Table 3): Please add the area changes for the two periods (at least the relative ones) including change rates per year.

P13 and 14/15 (Tables 4 and 5): as for Table 3, please add the area change (at lest in per cent) when the caption says the table is presenting area changes. Please also right adjust all numbers and check if the provided uncertainty values are necessary.

P14 (L7): As mentioned above, Kolka Glacier was basically removed from its bed in 2002 and regenerated afterwards (but not yet to its full size). Due to this special behaviour the glacier should be removed from all statistical analysis.

P14 (L10): … shown in Fig. 4b.

P15: Please consider using a bold font for the numbers in Fig. 4, they are partly difficult to see.

P16

L1-9: I suggest removing this highly speculative reasoning from the results section. It should also not be in the discussion as the statements are strange. For example, point c) indicates that the authors might not be fully aware how glaciers work. Why should geographic location (or altitude in a) have something to do with shrinkage rates? Glaciers are where they are because climate is as it is. As long as they have an accumulation area, it does not matter if climate is more continental or drier. They might be larger or smaller depending on mountain elevation and possibilities to accumulate snow but they will not shrink slower or faster due to their location. The only thing that would matter is when there are strong regional differences in climate change (such as locally increased precipitation). Please note that the most important control of locally averaged glacier area change rates is likely the size class distribution of the glaciers (please add), as relative area changes in general increase towards smaller glaciers (please add the related scatter plot). So area loss rates are normally higher where glaciers are smaller. For this reason only rates for glaciers in the same size class should be compared.

L4: Why citing here studies that are 35-50 years old?

L10: I would show the general characteristics of the glaciers in the study region before the changes are analysed. Please also add the size class distribution (by number and area) for different macro-regions.

L23: 'recession rates' is fine for area changes, but please use 'retreat rates' when referring to length changes.

L24: 26 and 28 years is quite a long period. Can it be excluded for all glaciers in the sample that advance phases in-between reduced the long-term mean value (see P17/L1)?

P17

L5: I think 'while' relates to time, it should thus be 'whereas' here.

L7 (Table 6): As for Table 3 and please align numbers right. To avoid confusion, I would not use terminus retreat but length change. Values should then be negative for retreat and positive for advance. What should terminus elevation tell us? Please consider providing glacier length here as there might be a relation between the two.

P18: Maybe the visibility of the scale bars can be improved?

P19/20 (Figs. 8 to 10, Tables 7 and 8)

As I think climate data cannot be directly related to area changes (see above), I do not need to have any of the figures and tables presented here. This also roots in the unreflected presentation of the data. They are shown but why? In particular the mean annual values presented in Fig. 8 and Table 7. What is their relation to the observed glacier changes? As a small point, I assume 'Mean monthly air temperature' is 'Mean annual air temperature' (and please right adjust all numbers)? As a comment to the graphics, I would recommend adding major tick marks also at the opposite site of each axis along with additional minor tick marks (one year / one degree step), temperature on the y-axis should be capitalized, and a space inserted before the ºC (and please do not use a zero in superscript for the º sign, this is a special symbol). It is also unclear to me why the trend lines are shown in Fig. 8 but not in Fig. 9, despite trends being much stronger for the summer months (according to Table 7)? Overall, it would be sufficient for me to just mention in the text that JJA temperatures (T) increased by about 0.7 to 1.2 degrees for the various climate stations. By also presenting the trends in precipitation (P) with increases from 10 to 30% it might be required to shortly explain how much P increase is required to compensate a 1

ºC increase in summer T. This might require performing a sensitivity study with a mass balance model. Just arguing that this increase was not sufficient because glaciers are in retreat (P21, L9) is not an explanation (in particular considering that some glaciers did not retreat). Bottom line, I would remove this entire climatic analysis as it gives rise to numerous questions that are not easy to solve, the relation with area changes is very weak, and there is actually no real analysis of these datasets.

L12/13: Why are mean annual values shown in Fig. 8 and Table 7 when the relation is only with JJA T? And what does 'consistent' mean? In particular, the mean annual values of Fig. 7 say little about related glacier changes.

P21

L2: 'clearly show': Where? There is neither a scatter plot of area change rates nor a figure illustrating this.

L4/5: See comments above: Stronger relative changes in regions with smaller glaciers occur because smaller glaciers show a larger decline in the mean (please check and add a related scatter plot). There is no need to introduce 'Jurasic sedimentary rocks' as an explanation.

L7-13: Please remove; this analysis makes no sense in my opinion (as described above). By concluding from 'suggests … mostly reflects influence of rising temperatures' that 'temperature was the main control on the early glacial fluctuations of the 21st century …' is strange. How can something that 'reflects an influence' be converted in the next sentence to the main control on the fluctuations? And why early 21 century, the 1960-1986 period should be the late 21st century? And why glacier fluctuations? Temperature is in general the variable being responsible for the long-term trends whereas fluctuations (retreat and advance on top of a general trend) are driven by shorter-term variability in precipitation. I stop here but the reminder of this section is also not good (e.g. what have the eastern Alps T trend over the 1929-2011 period to do with the Caucasus variability?).

L14-24: I think the comparison of area change rates does not work in this way. I do not understand why a comparison is performed with glaciers in Kamchatka, the Kodar mountains, the Canadian Rocky Mts. or the Andes? How do they relate to the study region? Or have they been selected because of the roughly similar 50-60 year period? What about intermediate advances during this period? Can they be excluded for all regions?

L17/22: Please give two decimals for the average rate of change per year. Over a 50-year period 0.5% per year can be anything from 22.5 to 27.5%.

L22: I would argue that compared to the Caucasus the glaciers in the Alps are comparably large. But I also notice that you here consider the effect of larger relative area losses for smaller glaciers. So the question is why this has not been considered before (L4/5)?

L24: Please always add a mean value per year. When periods differ, please also explain why they can be compared nevertheless.

L27: I do not understand this comparison. There could only be an underestimation when the 1960s UGI has an underestimation of glacier area in comparison to the 1911 PGI inventory. Has it? Compared to the more recent inventories the effect should be vice versa. I do also not understand why Khromova (2014) is saying that glacier decrease was faster in the first half of the 20th century, 24.7% in 70 years is much less than 17% in 40 years?

L30-33: Is it required to list here all river basins? Maybe it is more meaningful to write the total and percentage of area that has been missed?

L34: I fully agree on this, the country must be correctly stated in the attribute table. Apart from this, in the RGI Caucasus is only one first-order region.

L36ff: I would have liked to see these issues more prominently covered in the introduction as they provide a very good motivation to perform this study.

P22

L9-11: It would have been nice to see some of these issues illustrated in the study, also to improve consistency in interpretation by the science community. Maybe one or two examples can be added in the revised version?

L15/16: Again, please give two decimals for change rates per year.

L21: have retreated => have decreased in size (reserve advance/retreat for length changes).

L23/4: see comments above, I am sure that lithology does not play a role here.

L25: This might be correct, but there is not much evidence for this statement in the text before. Have area/volume change scenarios been calculated? Or maybe refer to one of the global scale studies that have done this.

L29-31: This conclusion is also not really based on a careful elaboration in the manuscript. Please also consider ice thickness distribution and glacier size as key factors impacting on area change rates. If values are compared across the region, please only compare glaciers in the same size class.

L32: 'may reduce these uncertainties': Which uncertainties? Please name them before.

**Additional References**

Bliss, A., Hock, R. and Radić, V. (2014), Global response of glacier runoff to twenty-first century climate change. J. Geophys. Res. Earth Surf., 119, 717-730, doi: 10.1002/2013JF002931.

Davies, B.J. and Glasser, N.F. (2012), Accelerating shrinkage of Patagonian glaciers from the Little Ice Age (~AD 1870) to 2011. J. Glaciol., 58(212), 1063-1084.

Fischer, A., et al. (2015), Tracing glacier changes in Austria from the Little Ice Age to the present using a lidar-based high-resolution glacier inventory in Austria. Cryosphere, 9, 753-766.

Gardner, A., et al. (2013), A reconciled estimate of glacier contributions to sea level rise: 2003 to 2009. Science, 340(6134), 852–857.

Guo, W.Q. and 10 others (2015), The second Chinese glacier inventory: data, methods and results. J. Glaciol., 61(226), 357–372.

Huss, M. (2012), Extrapolating glacier mass balance to the mountain-range scale: The European Alps 1900- 2100. Cryosphere, 6, 713-727

Kienholz, C., Herreid, S., Rich, J. L., Arendt, A. A., Hock, R., and Burgess, E. W. (2015), Derivation and analysis of a complete modern-date glacier inventory for Alaska and northwest Canada. J. Glaciol., 61, 403-420.

Nagai, H., Fujita, K., Sakai, A., Nuimura, T., and Tadono, T. (2016), Comparison of multiple glacier inventories with a new inventory derived from high-resolution ALOS imagery in the Bhutan Himalaya. Cryosphere, 10, 65-85.

Nuimura, T., et al. (2015), The GAMDAM glacier inventory: a quality-controlled inventory of Asian glaciers. Cryosphere, 9, 849-864.

Ojha, S. et al. (2016), Glacier area shrinkage in eastern Nepal Himalaya since 1992 using high-resolution inventories from aerial photographs and ALOS satellite images. J Glaciol., 62 (233), 512-524.

Paul, F. and Mölg, N. (2014), Hasty retreat of glaciers in northern Patagonia from 1985 to 2011. J. Glaciol., 60(224), 1033-1043.

Pope, A., Rees, W.G., Fox, A.J. and Fleming, A. (2014), Open access data in polar and cryospheric remote sensing. Remote Sensing, 6, 10.3390/ rs6076183.

Solomina, O., et al. (2016), Glacier variations in the Northern Caucasus compared to climatic reconstructions over the past millennium. Glob. Planet. Change, 140, 28-58.

Winsvold, S.H., Andreassen, L.M., and Kienholz, C. (2014), Glacier area and length changes in Norway from repeat inventories. Cryosphere, 8, 1885-1903.

Wulder M.A, et al. (2012), Opening the archive: how free data has enabled the science and monitoring promise of Landsat. Remote Sens. Environ., 122, 2-10.

Wulder, M.A., et al. (2016), The global Landsat archive: status, consolidation, and direction. Remote Sens. Environ., 185, 271-283.

---

## Referee Comment (RC2) · M Shahgedanova (Referee) · 28 Jun 2017

**The Greater Caucasus Glacier Inventory (Russia/Georgia/Azerbaijan)**

**L. Telidze and R.D. Wheate**

The paper by Telidze and Wheate presents an assessment of changes in areas of over 2000 glaciers and rates of glacier terminus retreat in the Caucasus Mountains using remote sensing and topographic maps. Two time steps 1960s-1980s and 1980s-2000s are used and comparisons with 1911 pop up at the end. Limited analysis of the detected changes is provided focusing on the north and south macroslopes, different river basins and, for terminus retreat, on aspect. Temperature and precipitation time series are presented providing context for the observed change.

The Caucasus Mountains and their glaciers are relatively well researched and there is a considerable amount of literature devoted to the regional glacier change (See Shahgedanova et al., 2014 for a reasonably comprehensive although by not exhaustive review). In comparison with the existing studies, and especially those published in English, this paper makes two not quite novel but useful contributions: (i) it assesses changes in comparison with the 1960s (unlike other recently published studies which mostly consider changes from the 1980s except for Khromova et al., 2009; 2014) and (ii) it examines changes in the extent of glaciers in the eastern Caucasus which is not fully covered by GLIMS and is missing from the recently published assessments. From the perspective of water resources, changes in the eastern Caucasus are important and much more so than in the western and central Caucasus because in the foothills, precipitation drops from about 2000 mm per year in the west to about 200 mm per year in the east. Detailed analysis what happens in this region would be valuable. Regrettably, the paper does not provide in-depth analysis building on the strengths of the data it generated and it is not clear what new contribution it makes in comparison with the earlier studies.

With regard to the data for the 1960s, derived from the topographic maps and the Catalogue of Glaciers of the USSR, the data presented here suffers from the problem faced by many other papers – there is no reliable accuracy assessment. Representation of glaciers on the 1:50000 maps is questionable as they were compiled by geodesists rather than glaciologists and issues of snow or debris cover were often neglected. The quality of data presented in the Catalogue also varies. So one can't just assign a 2.1% error to glacier areas as in the 1960s (p. 8; line 3); it is likely to be higher. The way to deal with this problem is either to re-map areas of a decent sample of glaciers of different size and type using aerial photographs or to use Corona instead. I don't know if aerial photographs are available in Georgia but some are available at the Institute of Geography, Russian Academy of Science with which the lead author is in contact. Corona can be obtained by anyone. This will be quite a lot of work but then the authors would have a clear idea of data quality and, in case of using Corona, will develop a new and valuable data set.

Another important issue with error analysis is that errors due to mapping by individual operators are not quantified. I suggest that glaciers of different sizes should be mapped by several operators to quantify the error.

In my view, the paper has a wrong balance between the introduction, description of study area and review of previous studies. I would place a review of the existing literature first and write it in a critical way showing which new or under-researched questions this paper addresses. I suggest that the authors should substantially cut the description of the study region as it contains a lot of information that is not directly relevant to the paper. There is no need to describe what is where, just show the western, central and eastern sectors on the map.

Surging glaciers are mentioned: have they been excluded from the analysis? There are a few particularly in the eastern sector (see Rototaeva et al., 2006 and a chapter on surging glaciers in the same volume). Either exclude or analyse as a separate group. If the published data are insuffiecient, maybe you can detect surging glaciers in the region?

Meteorological stations: The authors should exclude all those located in urban areas such as Vladikavkaz. The authors mention stronger temperature trends at this site which is most certainly due to an urban effect.

Debris cover: There are a substantial number of debris-covered glaciers in the region although not as much as in the south-eastern Asia. These are specifically described by Rototaeva et al (2006).

I suggest that these are analysed separately and some comparison with the recession of the debris-free glaciers is provided.

Specific comments

Tables 2 is too long and should be included as a Supplement at best. Can you present data on a map somehow?

Table 3: Show data for west-central-east as well as north and south. Changes in the eastern sector is your contribution so bring them out.

Tables 4 and 5: Again too much for the main text. It may be better to show changes in glacier areas rather than absolute values (which can go to a supplement). You comment about differences between the Elbrus and the Kazbek massifs but it is not easily seen from the tables which present glacier areas rather than change. If you show the changes in graphical format, it will illustrate your statements better.

P. 14, line 5: I would omit Kolka from this assessment; it's a catastrophic loss of ice which is not comparable with gradual area reduction.

You might want to comment more on the differences between the rates of glacier area reduction in the Elbrus and Kazbek massifs. While there were several publications about the Elbrus (Shahgedanova et al 2014; Holobaka 2013; Zolotarev and Kharkovets, 2007; 2012) to which your data do not add much, less is written about the Kazbek so you may want to explore the data further. Again, this will highlight west-east gradient in glacier area reduction.

Retreat of glacier termini: Why only 14 glaciers? This does not give you very good statistics.

P. 21 Line 5: Why is geology important?

P. 21 Lines 10-15: Not all of your comparisons make sense. Glaciers in the European Alps are much larger, in the Kodar they are much smaller and these are cold-based glaciers. In Kamchatka, they are altogether different and affected by volcanic activity. Looks like a random selection of papers for comparison.

P. 22 Line 20: I am not sure how geology affects faster loss of glacierized area in the east. I also can't see the difference in the warming rates between west and east (especially given that the only really high-altitude station, Klukhorsky Pereval, is in the west). It is more likely to be an effect of (i) drier climate in the east especially in comparison with the very humid western sector; (ii) lower elevations in comparison with the central sector) and (iii) size and type of glaciers prevailing in the regions. That's were insufficient data analysis shows: you need to analyse your changes in glacierized area according

to glacier type, size and elevation and compare these categories for western central and eastern sectors as well as north and south (to account for the influence of the North Atlantic Oscillation).

References

Holobaca, I.-H: Glacier Mapper – a new method designed to assess change in mountain glaciers, Int. J. Remote Sens., 34, 8475–8490, 2013.
Khromova, T., Nosenko, G. and Chernova L.: Mapping of glacier extent changes in the mountain regions using space images
and glacier inventories, the 24th International Cartographic Conference, Santiago, Chile, 2009.
Khromova, T., Nosenko, G., Kutuzov, S., Muraviev and, A., and Chernova, L.: Glacier area changes in Northern Eurasia, Environ. Res. Lett., 9, 015003, doi:10.1088/1748-9326/9/1/015003, 2014.
Rototaeva, O., Nosenko, G., Tarasova, L., and Khmelevsky, I.: Caucasus, in: Oledenenie Severnoi i Tsentralnoi Evrazii v Sovremennuyu Epohu (Glaciation in N orthern and Central Eurasia at Present Time), edited by: Kotlyakov, V. M., Nauka Publishers, Moscow, 141–162, 2006 (in Russian).
Shahgedanova, M., Nosenko, G., Kutuzov, S., Rototaeva, O., and Khromova, T.: Deglaciation of the Caucasus Mountains, Russia/Georgia, in the 21st century observed with ASTER satellite imagery and aerial photography, The Cryosphere, 8, 30 2367–2379, doi:10.5194/tc-8-2367-2014, 2014.
Zolotarev, E. A. and Kharkovets, E. G.: Oledenenie Elbrusa kontse XX veka (Glaciation of Mt. Elbrus at the end of the 20th Century), Led i Sneg (Ice and Snow), 5, 45–51, 2007 (in Russian).
Zolotarev, E. A. and Kharkovets, E. G.: Evolyutsiya oledeneniya Elbrusa posle malogo lednikovogo perioda (Evolution of glaciation on Mt Elbrus after the Little Ice Age), Led i Sneg (Ice and Snow), 2, 5–14, 2012 (in Russian).

---

## Editor Comment (EC1) · C. R. Stokes (Editor) · 28 Jun 2017

Dear Levan,

You will, by now, have received formal notification that the open discussion of your manuscript has closed.

I would like to thank the two reviewers (Frank Paul and Maria Shahgedanova) for their very helpful comments on your manuscript. Whilst they both see the clear potential of your work, they identify a large number of important issues that need to be addressed and which would constitute major revisions and a re-review of your manuscript.

In particular, I agree that you will need to provide a much stronger rationale for the study, given that there are several similar papers from the region, including many by the

lead author. This should summarise previous work and, importantly, point out where the gaps are that still need to be addressed and why. I also agree that you need to provide a more thorough explanation of errors and uncertainties, as raised by both reviewers. Perhaps the most important issue, however, is that the analysis of your results will need to be much more detailed and substantive in order to increase the wider significance of your manuscript. In addition, both reviewers have helpfully identified a large number of specific and more minor issues that you should carefully address if you decide to re-submit your manuscript.

If you have any queries, please do not hesitate to get in touch.

Kind regards,

Chris Stokes Editor

---

## Author Comment (AC1) · 17 Jul 2017

Dear Dr. Chris Stokes

We would first like to thank you for submitting our manuscript in TCD and giving chance to improve our manuscript.

Although there are several previous studies about the Greater Caucasus, we decide re-submit the manuscript as we present here three complete inventory covering a larger area than earlier studies. We provide all correction according the referees suggestions. Please see the revised manuscript.

Kind regards

---

## Author Comment (AC2) · 17 Jul 2017

**Authors reply to Dr. Frank Paul's comments**

**"The Greater Caucasus Glacier Inventory (Russia/Georgia/Azerbaijan)" by L. G. Tielidze and R. D. Wheate**

| |
|---|
| *P1* |
| *L18/19: 'marked acceleration': This is not obvious from the presented numbers. Please add relative change rates per year for both periods (with two decimals!)* |
| We agree, please see P - 1. L – 22-24. |
| *L22: 'can be used': have/will the outlines and inventory data be submitted to GLIMS? If yes, please add so that it is clear they are accessible.* |
| We agree, please see P 1. L – 24-25. |
| *L27: The Fischer et al. (2015) study is maybe not so relevant for global sea level rise. I suggest writing 'for local to regional-scale hydrological studies (Fischer et al. 2015, Huss 2012), to global calculation of sea level change (Bliss et al. 2014, Gardner et al. 2013).* |
| We agree, please see P 1. L – 28-30. |
| *L27-31: I do not understand this sentence, please rewrite. Please note (this still goes wrong in many studies), there is no direct link between climate change and change in glacier area. Changes in area result from the combined interaction of surface lowering (which is a result of glacier mass balance) and ice thickness distribution. So for the same climatic forcing (at the same elevation) area changes can be very large when the ice is thin or small when the ice is thick. This has little to do with response times or the geometric adjustment of a glacier on the decadal time scale.* |
| We agree, please see P 1. L – 30-32. |
| *L32: Tracking only the area changes over many decades might not help much. A good under-standing also requires to have surface elevation changes and ice thickness distribution.* |
| We deleted this sentence. |
| *L34: Maybe the cited studies are not the best examples. Recently published glacier inventories often cover entire mountain ranges with thousands to ten-thousands of glaciers (e.g. for Alaska, Greenland, the Alps, or the Greater Himalaya).* |
| We deleted this sentence. |
| *L36: I think this sentence needs to have a connection to the sentence before (e.g. 'However, consistent methodological inventories are necessary to correctly perform change assess-ment and other glaciological research …'). Unfortunately, this point is not taken up again to motivate (or even justify) the inventory presented here (see next point).* |
| We deleted this sentence. |
| *P2* |
| *L7: Yes, glacier research certainly is. But why is it required to have yet another inventory of the region? Apart from several general statements, the introduction makes no attempt to motivate what is presented afterwards. When readers should also read the rest of the text, it must be clearly explained in the introduction what the research gaps are (for this region) and how this study is addressing them and presenting at the same time a never seen before analysis of the new dataset that has only become possible now. There is now a large op- portunity to do so, so please do it!* |
| We agree, please see P 1. L – 40-45 and P 2. L – 1-10. |
| *L20ff: I think the text before is fine to introduce the study region. But from here until P3/L28 I suggest removing the text as the contents (descriptions of mountains and their height/location) is not used later in the text and digitally available for most of us.* |
| We agree, please see P 2-3. "**2 Study Area**" section. |

| |
|---|
| *P2* |
| *L25 (Fig. 1): I am not convinced that the presentation of area changes in sub-catchments (Table 2) is really required. If the authors agree that they are not required, I would remove the catchment numbers here (but please provide them as a separate shape file with the dataset) and show instead where the glaciers depicted in Figs. 2 to 7 are located.* |

We changed Fig. 1 and moved Table 2 in the Supplement. We think the sub-catchments is required.

| |
|---|
| *P3* |
| *L29ff: Some climatic background is certainly fine, but how does the information provided here help me to understand the results of the inventory, e.g. for what purpose do I need to know the minimum and maximum lapse rates or the minimum air temperature in 1983? What I want to know is mean summer or annual temperature and precipitation at the mean elevation of glaciers.* |

We agree and deleted all these sentences.

| |
|---|
| *P4* |
| *L5/6: LIA history: I suggest adding here the study by Solomina et al. (2016).* |

We agree, please see P 2. L – 30.

| |
|---|
| *L14ff: Where does the ELA come from? ELA is related to glacier mass balance and the gradients provided here suggest that many glaciers with mass balance measurements have been used to derive it (which ones?). Note: snow lines or glacier mean elevation ≠ ELA.* |

We deleted this sentence.

| |
|---|
| *L20: Please use 'retreat/advance' only when reference is made to changes in length. For mass balance one can use mass loss/gain.* |

We agree, please see P 3. L – 7.

| |
|---|
| *L39: 786 glaciers* |

We agree, please see P 3. L – 28.

| |
|---|
| *P5* |
| *L12/13: Please give area changes always in % as absolute values are incomparable. Moreover, a comparison of area change over periods of different length should always provide the rate (i.e. per cent per year).* |

We agree, please see P 3. L – 32-40.

| |
|---|
| *L20: Fig. 2d/e: Please do some contrast enhancement here, there is not much to see.* |

We changed this figure, please see P 12. Fig. 9.

| |
|---|
| *P6* |
| *L3/4: 'retreated most dramatically': What does this mean? Is it really retreat (length change) or area change? Please provide values for such statements.* |

We clarified, please see P 4. L – 6-7.

| |
|---|
| *L6: Here are all the salami-slices I referred to in the general comments. Please use this publication to make some progress in the analysis. There is no need to do the same paper again and again.* |

We deleted all salami-slices.

| |
|---|
| *L17: These references are ok but maybe somewhat out-dated? What about Wulder et al. (2012 / 2016) or Pope et al. (2014)?* |

We agree, please see P 4. L – 15.

| |
|---|
| *L26: 'cast an obvious shadow': The problem with retreating glaciers is that their terminus does no longer 'cast an obvious shadow' and that most glaciers in the world are in retreat. So what has been done when their terminus is barely visible?* |

Please see P 4. L – 26.

| |
|---|
| *L26: manual delineation: this is correct but for clean ice automated mapping is superior (consistent and reproducible), at least in the absence of seasonal snow* |

| |
|---|
| We agree, please see P 5. L – 12-16. |
| *L28: Not only glacier tongues need to be free of seasonal snow, the entire glacier should be.* |
| We agree, please see P 4. L – 28. |
| *L34: 'the most accurate method': this is certainly correct for debris-covered glaciers, but for clean ice automated mapping has the same accuracy (and outlines are consistent, not generalized and reproducible; all important assets these days).* |
| We agree, please see P 5. L – 12-16. |
| *L37: of 20 metres from 88 aerial (remove brackets)* |
| We agree, please see P 4. L – 33. |
| *P7*
 *L12: I assume this should read 'To estimate uncertainty of glacier area'? Please check carefully the difference between uncertainty (of a measurement) and error (difference to a reference dataset). These are different. For example (L14), the 'digitizing error' should likely read 'digitizing uncertainty'.* |
| We agree, please see P 6. L – 11-13. |
| *L14: Please clarify how the map rectification could have an impact on the derived glacier area when considering this as an 'error term'. As far as I know, geolocation uncertainty is only an issue when directly calculated from different datasets (e.g. cumulative length changes using digital intersection). Location errors should not impact on the derived area.* |
| We agree, please see P 6. L – 13-14. |
| *L16: I think all these are not errors but uncertainties.* |
| We agree, please see P 6. L – 11, 13, 14 and 18. |
| *L18: You can see in Paul et al. (2013) how line-placing uncertainty for manual digitizing looks like. It is indeed +/-1 pixel (and worse for higher resolution imagery due to increasing generalization). A more realistic result can thus be achieved when using a +/-1 pixel buffer (and +/-2 pixels for debris-covered glaciers).* |
| We agree, please see P 6. L – 18-30 and P 7. L – 1-3, |
| *P8*
 *L4: conservative: It is foremost just a statistical value with probably little relation to the analysts work. I would thus strongly recommend to perform a multiple (3-5 times) digitizing experiment with a couple (about 10) of differently sized glaciers (with debris and clean). This should be done for all three datasets (map, TM, ASTER) and provide a more realistic estimate of the uncertainty.* |
| We agree, please see P 5-6, "**3.2 Glacier error assessment**" section |
| *L5: Figs. 3a and b should be shown side by side. It might also e helpful to mark the boundary of Kolka Glacier in Fig. 3b as the 64% debris cover mentioned in L15 cannot be seen.* |
| We deleted this figure as you recommended bellow. |
| *L11: This seems to be a repetition from P6/L34.* |
| We agree and deleted this sentence. |
| *L14: Why not adding that Kolka has rebuild from ice avalanches from the surrounding steep rock walls after being completely removed from its bed by the 2002 avalanche? This might help to explain why it is such a special case.* |
| We agree, please see P 9. L – 8-9. |
| *L17: 'not typical': why? I think the Caucasus has quite a lot of debris-covered glaciers, in particular the larger ones.* |
| We deleted this sentence. |
| *P9*
 *L8: As mentioned above, there is no direct link between glacier area change and climate. When climate data are analysed for the same time period you have to analyse mass* |

*balance data, only these provide a direct and undelayed response to the governing atmospheric conditions. Also for a general trend analysis the forcing has to consider glacier response times. Understanding glacier area changes for 1960 to 1986 might thus require looking at climatic conditions from 1930-1960 (for an assumed 30 year response time). But even this might only allow explaining general trends in \*length\* changes as area changes are also driven by the ice thickness distribution. I think it is ok to say that glaciers have lost area since the 1960s because temperatures have increased, but that's it.*

We deleted all climatic data as you suggest bellow.

*L13/15 and elsewhere: please always provide relative area change rates (per year) to have comparable values. And please always give two decimals! The rounded values in L17, 19, 20, 22, and 24 are not accurate enough (see example below).*

We agree, please see P 7. L – 12-20; P 8. L – 1-28; P 9. L – 1-11;

*L27: 'highest glacier surface decrease': What is this? Absolute loss, relative change, change rate? Please be precise.*

We clarified, please see P 8. L – 17.

*P10-12 (Table 2): Please rethink if this Table makes any sense. I suggest removing it completely and refer to the digital datasets for such assessments. If it should stay, move it to the Appendix and better justify why these numbers per tributary river basin are required. There is currently no further use of them. Instead of providing (statistical) uncertainties for each basin, please consider adding absolute/relative area changes. This could also be visualized graphically for all regions (using a multi-segmented bar chart) or in dependence of elevation. Please also note that numbers should be aligned right rather than left.*

We agree, please see Supplement, P 1. Table 1. Also in the manuscript Fig. 5-7.

*P12 (Table 3): Please add the area changes for the two periods (at least the relative ones) including change rates per year.*

We agree, please see P 8. Table 2.

*P13 and 14/15 (Tables 4 and 5): as for Table 3, please add the area change (at lest in per cent) when the caption says the table is presenting area changes. Please also right adjust all numbers and check if the provided uncertainty values are necessary.*

We agree, please see Supplement, P 4-7. Table 3-4.

*P14 (L7): As mentioned above, Kolka Glacier was basically removed from its bed in 2002 and regenerated afterwards (but not yet to its full size). Due to this special behaviour the glacier should be removed from all statistical analysis.*

We agree, P 9. L – 8-10.

*P15: Please consider using a bold font for the numbers in Fig. 4, they are partly difficult to see.*

We agree, please see Supplement Fig. 1-2

*P16*

*L1-9: I suggest removing this highly speculative reasoning from the results section. It should also not be in the discussion as the statements are strange. For example, point c) indicates that the authors might not be fully aware how glaciers work. Why should geographic loca- tion (or altitude in a) have something to do with shrinkage rates? Glaciers are where they are because climate is as it is. As long as they have an accumulation area, it does not mat- ter if climate is more continental or drier. They might be larger or smaller depending on mountain elevation and possibilities to accumulate snow but they will not shrink slower or faster due to their location. The only thing that would matter is when there are strong regional differences in climate change (such as locally increased precipitation). Please note that the most important control of locally averaged glacier area change rates is likely the size class distribution of the glaciers (please add), as relative area changes in general increase towards smaller glaciers*

| |
|---|
| *(please add the related scatter plot). So area loss rates are normally higher where glaciers are smaller. For this reason only rates for glaciers in the same size class should be compared.* |
| We agree and removed this paragraph. |
| *L4: Why citing here studies that are 35-50 years old?* |
| We agree and removed this citations. |
| *L10: I would show the general characteristics of the glaciers in the study region before the changes are analysed. Please also add the size class distribution (by number and area) for different macro-regions.* |
| We agree, please see P 9-10. "**4.2 Glacier characteristics**" section |
| *L23: 'recession rates' is fine for area changes, but please use 'retreat rates' when referring to length changes.* |
| We agree, P 11. L – 6. |
| *P17*
 *L5: I think 'while' relates to time, it should thus be 'whereas' here.* |
| We agree, P 9. L – 36. |
| *L7 (Table 6): As for Table 3 and please align numbers right. To avoid confusion, I would not use terminus retreat but length change. Values should then be negative for retreat and positive for advance. What should terminus elevation tell us? Please consider providing glacier length here as there might be a relation between the two.* |
| We agree, please see Supplement P. 13-14. Table 5. |
| *P18: Maybe the visibility of the scale bars can be improved?* |
| We agree please see Supplement P. 15. Fig. 14. |
| *P19/20 (Figs. 8 to 10, Tables 7 and 8) As I think climate data cannot be directly related to area changes (see above), I do not need to have any of the figures and tables presented here. This also roots in the unreflected presentation of the data. They are shown but why? In particular the mean annual values presented in Fig. 8 and Table 7. What is their relation to the observed glacier changes? As a small point, I assume 'Mean monthly air temperature' is 'Mean annual air temperature' (and please right adjust all numbers)? As a comment to the graphics, I would recommend adding major tick marks also at the opposite site of each axis along with additional minor tick marks (one year / one degree step), temperature on the y-axis should be capitalized, and a space inserted before the °C (and please do not use a zero in superscript for the ° sign, this is a special symbol). It is also unclear to me why the trend lines are shown in Fig. 8 but not in Fig. 9, despite trends being much stronger for the summer months (according to Table 7)? Overall, it would be sufficient for me to just mention in the text that JJA temperatures (T) increased by about 0.7 to 1.2 degrees for the various climate stations. By also presenting the trends in precipitation (P) with increases from 10 to 30% it might be required to shortly explain how much P increase is required to compensate a 1 °C increase in summer T. This might require performing a sensitivity study with a mass balance model. Just arguing that this increase was not sufficient because glaciers are in re- treat (P21, L9) is not an explanation (in particular considering that some glaciers did not retreat). Bottom line, I would remove this entire climatic analysis as it gives rise to numerous questions that are not easy to solve, the relation with area changes is very weak, and there is actually no real analysis of these datasets.* |
| We agree and removed entire climatic |
| *P21*
 *L2: 'clearly show': Where? There is neither a scatter plot of area change rates nor a figure illustrating this.* |
| We changed this sentence, please see P 11. L – 37-38; P 12 and P 10. Fig. 6. |
| *L4/5: See comments above: Stronger relative changes in regions with smaller glaciers occur because smaller glaciers show a larger decline in the mean (please check and add a* |

| | |
|---|---|
| *related scatter plot). There is no need to introduce 'Jurasic sedimentary rocks' as an explanation.* | |
| We changed this sentence, please see P 11. L – 37-38; P 12 and P 10. Fig. 6. | |
| *L7-13: Please remove; this analysis makes no sense in my opinion (as described above). By concluding from 'suggests … mostly reflects influence of rising temperatures' that 'temperature was the main control on the early glacial fluctuations of the 21st century …' is strange. How can something that 'reflects an influence' be converted in the next sentence to the main control on the fluctuations? And why early 21 century, the 1960-1986 period should be the late 21st century? And why glacier fluctuations? Temperature is in general the variable being responsible for the long-term trends whereas fluctuations (retreat and advance on top of a general trend) are driven by shorter-term variability in precipitation. I stop here but the reminder of this section is also not good (e.g. what have the eastern Alps T trend over the 1929-2011 period to do with the Caucasus variability?).* | |
| We agree and removed according your suggestion. | |
| *L14-24: I think the comparison of area change rates does not work in this way. I do not un-derstand why a comparison is performed with glaciers in Kamchatka, the Kodar moun-tains, the Canadian Rocky Mts. or the Andes? How do they relate to the study region? Or have they been selected because of the roughly similar 50-60 year period? What about in- termediate advances during this period? Can they be excluded for all regions?* | |
| Please see new "**5 Discussion**" section P 11-12. | |
| *L27: I do not understand this comparison. There could only be an underestimation when the 1960s UGI has an underestimation of glacier area in comparison to the 1911 PGI inventory. Has it? Compared to the more recent inventories the effect should be vice versa. I do also not understand why Khromova (2014) is saying that glacier decrease was faster in the first half of the 20th century, 24.7% in 70 years is much less than 17% in 40 years?* | |
| We clarified. Please new comparison, P 11. L – 29-36. | |
| *L30-33: Is it required to list here all river basins? Maybe it is more meaningful to write the total and percentage of area that has been missed?* | |
| We think, should be mentioned here all missed river basins. | |
| *L36ff: I would have liked to see these issues more prominently covered in the introduction as they provide a very good motivation to perform this study.* | |
| We agree, please see P 12. L – 43-45 and P 2. L – 1-9. | |
| *P22* *L9-11: It would have been nice to see some of these issues illustrated in the study, also to improve consistency in interpretation by the science community. Maybe one or two examples can be added in the revised version?* | |
| We agree, please see P 6-7. Fig. 2-3. | |
| *L15/16: Again, please give two decimals for change rates per year.* | |
| We agree, please see P 13. L 15-24. | |
| *L21: have retreated => have decreased in size (reserve advance/retreat for length changes).* | |
| We agree, please see P 13. L 18. | |
| *L23/4: see comments above, I am sure that lithology does not play a role here.* | |
| We agree and deleted this sentence. | |
| *L25: This might be correct, but there is not much evidence for this statement in the text be-fore. Have area/volume change scenarios been calculated? Or maybe refer to one of the global scale studies that have done this.* | |
| We agree and refer suitable reference in the text before, please see P 8. L 17-19. | |
| *L29-31: This conclusion is also not really based on a careful elaboration in the manuscript. Please also consider ice thickness distribution and glacier size as key factors impacting on area change rates. If values are compared across the region, please only compare* | |

| |
|---|
| *glaciers in the same size class.* |
| We agree, please see new "**5 Discussion**" and "**6 Conclusions**" senctions |
| *L32: 'may reduce these uncertainties': Which uncertainties? Please name them before.* |
| We deleted this sentence. |

---

## Author Comment (AC3) · 17 Jul 2017

**Authors reply to Dr. Maria Shahgedanova's comments**

**"The Greater Caucasus Glacier Inventory (Russia/Georgia/Azerbaijan)" by L. G. Tielidze and R. D. Wheate**

| |
|---|
| *With regard to the data for the 1960s, derived from the topographic maps and the Catalogue of Glaciers of the USSR, the data presented here suffers from the problem faced by many other papers – there is no reliable accuracy assessment. Representation of glaciers on the 1:50000 maps is questionable as they were compiled by geodesists rather than glaciologists and issues of snow or debris cover were often neglected. The quality of data presented in the Catalogue also varies. So one can't just assign a 2.1% error to glacier areas as in the 1960s (p. 8; line 3); it is likely to be higher. The way to deal with this problem is either to re-map areas of a decent sample of glaciers of different size and type using aerial photographs or to use Corona instead.* |
| We agree there wasn't reliable accuracy assessment before, but we don't think that 1:50000 maps are questionable as they were created by the best specialists for that time. Also, creation of the topographical maps is Geodesists job and not glaciologists and this does not stop us using their data. In addition, these maps are the only source nowadays that completely cover the entire Caucasus mountains, as the Corona and survived original imagery covers just small individual sections. Also the large scale topographical maps from 20th century are widely used for glacier assessment and inventories (Granshaw and Fountain, 2006; Andreassen, et al., 2008; Tennant et al., 2012; Tennant and Menounos, 2013; Winsvold et al., 2014; and many others). The USSR glaciers catalog is also often used for the former USSR glacier comparison with modern dataset (Khromova et al., 2014; Kotlyakov et al., 2015; Lynch et al., 2016 etc..). As for the uncertainty, after manuscript revision, our uncertainty increased over the ±4.4%. |
| *Another important issue with error analysis is that errors due to mapping by individual operators are not quantified. I suggest that glaciers of different sizes should be mapped by several operators to quantify the error.* |
| We agree. Please see P. 5 - "**3.2 Glacier error assessment**" section. |
| *In my view, the paper has a wrong balance between the introduction, description of study area and review of previous studies. I would place a review of the existing literature first and write it in a critical way showing which new or under-researched questions this paper addresses. I suggest that the authors should substantially cut the description of the study region as it contains a lot of information that is not directly relevant to the paper.  There is no need to describe what is where, just show the western, central and eastern sectors on the map.* |
| We agree, please see P 1-4. We changed first three section and Fig. 1. |
| *Meteorological stations: The authors should exclude all those located in urban areas such as Vladikavkaz. The authors mention stronger temperature trends at this site which is most certainly due to an urban effect.* |
| According the Dr. F Paul suggestion, we deleted all meteorological data. |
| *Tables 2 is too long and should be included as a Supplement at best. Can you present data on a map somehow?* |
| We agree, Please see P. 2 - Fig. 1 and Supplement - Table 1. |
| *Table 3: Show data for west-central-east as well as north and south. Changes in the eastern sector is your contribution so bring them out.* |
| We agree, please see P. 7-8 – "**4.1 Area and number change**" section, Table 3 and Fig. 4. |
| *Tables 4 and 5: Again too much for the main text. It may be better to show changes in glacier areas rather than absolute values (which can go to a supplement). You comment about differences between the Elbrus and the Kazbek massifs but it is not easily seen from the tables which present glacier areas rather than change. If you show the changes in graphical* |

| |
|---|
| *format, it will illustrate your statements better.* |
| We agree, please see P. 8 – Fig. 4, and Supplement P. 4-7 – Table – 3-4 and Fig.  1-2 |
| *P. 14, line 5: I would omit Kolka from this assessment; it's a catastrophic loss of ice which is not comparable with gradual area reduction.* |
| We agree, please see P. 9, L. 8-10 |
| *Retreat of glacier termini: Why only 14 glaciers? This does not give you very good statistics.* |
| We agree. We chose 30 glaciers with four different classes. Please see P. 11 "**4.3 Glacier length change**" section and Supplement – Table 5. |
| *P. 21 Line 5: Why is geology important?* |
| We changed this sentence, please see P. 11-12. L. 37-1 |
| *P. 21 Lines 10-15: Not all of your comparisons make sense. Glaciers in the European Alps are much larger, in the Kodar they are much smaller and these are cold-based glaciers. In Kamchatka, they are altogether different and affected by volcanic activity. Looks like a random selection of papers for comparison.* |
| We agree. We deleted this paragraph. Please see P. 11-12, "**5 Discussion**" section |
| *P. 22 Line 20: I am not sure how geology affects faster loss of glacierized area in the east.  I also can't see the difference in the warming rates between west and east (especially given that the only really high-altitude station, Klukhorsky Pereval, is in the west). It is more likely to be an effect of (i) drier climate in the east especially in comparison with the very humid western sector; (ii) lower elevations in comparison with the central sector) and (iii) size and type of glaciers prevailing in the regions. That's were insufficient data analysis shows: you need to analyse your changes in glacierized area according to glacier type, size and elevation and compare these categories for western central and eastern sectors as well as north and south (to account for the influence of the North Atlantic Oscillation).* |
| We deleted sentence about the "geology affect", Please see P. 13, "**6 Conclusions**" section. Also, as we already mentioned, we deleted all meteorological data according the Dr. F. Paul suggestion. In addition, for glacier characteristic we added new section. Please see P. 9-10 "**4.2 Glacier characteristics**" |

**References**

Andreassen, L. M., Paul, F., Kääb, A., and Hausberg, J. E.: Landsat-derived glacier inventory for Jotunheimen, Norway, and deduced glacier changes since the 1930s, The Cryosphere, 2, 131-145, https://doi.org/10.5194/tc-2-131-2008, 2008.

Granshaw, F. D. and Fountain, A. G.: Glacier change (1958–1998) in the North Cascades National Park Complex, Washington, USA, J. Glaciol., 52, 251–256, doi:10.3189/172756506781828782, 2006.

Khromova, T., Nosenko, G., Kutuzov, S., Muraviev and, A., and Chernova, L.: Glacier area changes in Northern Eurasia, Environ. Res. Lett., 9, 015003, doi:10.1088/1748-9326/9/1/015003, 2014.

Kotlyakov, V. M., Khromova, T. E., Nosenko, G. A., Popova, V. V., Chernova, L. P., and Murav'ev A. Ya.: New Data on Current Changes in the Mountain Glaciers of Russia, Doklady Earth Sciences, Vol. 464, Part 2, pp. 1094–1100. DOI: 10.1134/S1028334X15100207, 2015.

Lynch, C. M., Barr, I. D., Mullan, D., and Ruffell, A.: Rapid glacial retreat on the Kamchatka Peninsula during the early 21[st] century, The Cryosphere, 10, 1809-1821, doi:10.5194/tc-10-1809-2016, 2016.

Tennant, C., Menounos, B., Wheate, R., and Clague, J. J.: Area change of glaciers in the Canadian Rocky Mountains, 1919 to 2006, The Cryosphere, 6, 1541-1552, https://doi.org/10.5194/tc-6-1541-2012, 2012.

Tennant, C., and Menounos, B.: Glacier change of the Columbia Icefield, Canadian Rocky Mountains, 1919-2009. Journal of Glaciology, vol. 59, issue 216, pp. 671-686, 2013.

Winsvold, S. H., Andreassen, L. M., and Kienholz, C.: Glacier area and length changes in Norway from repeat inventories, The Cryosphere, 8, 1885-1903, https://doi.org/10.5194/tc-8-1885-2014, 2014.

---

## Author Response (AR1)

**Response letter for reviewer and editor comments**

**"The Greater Caucasus Glacier Inventory (Russia/Georgia/Azerbaijan)" by L. G. Tielidze and R. D. Wheate**

http://www.the-cryosphere-discuss.net/tc-2017-48/
doi.org/10.5194/tc-2017-48

Dear referees and editor,

We have took your remaining comments and checked our manuscript very carefully.

We appreciate the support and constructive reviews of editor and both referees.

All correction and change what we did is yellow.

Thank you

[revised manuscript text omitted]

\* Omitted in WGI database

[Figure]

**Figure 1.** Changes in glacierized area of Elbrus between 1960-1986-2014. See Tables 3 for the change statistics of individual glaciers. The 3/08/2014 Landsat 8 images was used as background.

**Table 4.** Kazbegi-Dzhimara massif glacier number and area change in 1960-1986-2014. All glaciers are shown in Fig. 2.

| | Kazbegi-Dzhimara massif glaciers | | Topographic maps 1960 | Landsat 5, 06/08/1986 | Landsat 8, 28/08/14/ | Area change | | |
|---|---|---|---|---|---|---|---|---|
| # | Name | WGI ID | Area km$^2$ | Area km$^2$ | Area km$^2$ | 1960-1986 % yr$^{-1}$ | 1986-2014 % yr$^{-1}$ | 1960-2014 % yr$^{-1}$ |
| 1 | Mydagrabyn | SU4G08010031 | 9.98±0.22 | 9.73±0.24 | 8.16±0.24 | -0.09 | -0.57 | -0.33 |
| 2 | Unnamed* | Unknown | - | - | 0.16±0.02 | 0 | 0 | 0 |
| 3 | Kolka | SU4G08010039 | 5.06±0.7 | 4.28±0.48 | 2.51±0.44 | -0.59 | -1.47 | -0.91 |
| 4 | Unnamed | Unknown | - | - | 0.75±0.04 | 0 | 0 | 0 |
| 5 | Unnamed | SU4G08010040 | 0.79±0.06 | 0.73±0.04 | 0.50±0.04 | -0.29 | -1.12 | -0.68 |
| 6 | Maili | SU4G08010041 | 7.29±0.14 | 6.75±0.14 | 6.57±0.12 | -0.28 | -0.09 | -0.18 |
| 7 | Unnamed | Unknown | - | - | 0.05±0.008 | 0 | 0 | 0 |
| 8 | Unnamed | SU4G08010042 | - | 0.67±0.04 | 0.44±0.02 | 0 | -1.22 | 0 |
| 9 | Chachi | SU4G08011046 | 2.61±0.06 | 2.57±0.08 | 1.85±0.08 | -0.05 | -1.00 | -0.53 |
| 10 | Unnamed | Unknown | - | - | 0.08±0.012 | 0 | 0 | 0 |
| 11 | Unnamed | SU4G08011047 | 0.86±0.06 | 0.52±0.04 | 0.12±0.02 | -1.52 | -2.74 | -1.59 |
| 12 | Unnamed | Unknown | - | - | 0.05±0.008 | 0 | 0 | 0 |
| 13 | Devdoraki | SU4G08011048 | 7.19±0.2 | 6.96±0.2 | 4.40±0.12 | -0.12 | -1.31 | -0.71 |
| 14 | Unnamed | Unknown | - | - | 1.78±0.08 | 0 | 0 | 0 |
| 15 | Abano | SU4G08011049 | 1.96±0.08 | 1.49±0.08 | 1.33±0.1 | -0.92 | -0.38 | -0.60 |
| 16 | Unnamed | Unknown | - | - | 0.03±0.004 | 0 | 0 | 0 |
| 17 | Unnamed | Unknown | 0.58±0.04 | 0.34±0.04 | 0.1±0.004 | -1.59 | -2.52 | -1.53 |
| 18 | Gergeti | SU4G08011052 | 6.82±0.18 | 6.26±0.2 | 5.77±0.18 | -0.31 | -0.27 | -0.28 |
| 19 | None | SU4G08011056 | 0.49±0.04 | 0.39±0.02 | 0.36±0.02 | -0.78 | -0.27 | -0.49 |
| 20 | Denkara | SU4G08011057 | 1.33±0.04 | 0.93±0.04 | 0.31±0.02 | -1.15 | -2.38 | -1.42 |
| 21 | Unnamed | Unknown | 0.49±0.04 | 0.06±0.01 | 0.03±0.004 | -3.37 | -1.78 | -1.73 |
| 22 | Unnamed | SU4G08011058 | 0.89±0.06 | 0.63±0.04 | 0.53±0.04 | -1.12 | -0.56 | -0.74 |
| 23 | Unnamed | SU4G08011059 | 1.12±0.04 | 0.98±0.06 | 0.75±0.06 | -0.48 | -0.83 | -0.61 |
| 24 | Mna | SU4G08011060 | 3.25±0.1 | 2.89±0.12 | 2.59±0.12 | -0.42 | -0.37 | -0.37 |
| 25 | Unnamed | SU4G08011061 | 1.57±0.04 | 1.30±0.06 | 1.44±0.04 | -0.66 | +0.38 | -0.15 |

| 26 | Suatisi Eastern | SU4G08011062 | 10.84±0.2 | 9.87±0.24 | 8.89±0.18 | -0.34 | -0.35 | -0.33 |
|---|---|---|---|---|---|---|---|---|
| 27 | Unnamed | Unknown | - | - | 0.08±0.012 | 0 | 0 | 0 |
| 28 | Suatisi Central | SU4G08011063 | 2.62±0.1 | 2.32±0.08 | 2.07±0.08 | -0.44 | -0.38 | -0.39 |
| 29 | Unnamed | Unknown | - | 0.29±0.04 | 0.23±0.02 | 0 | -0.73 | 0 |
| 30 | Suatisi Western | SU4G08011064 | 2.49±0.08 | 2.16±0.08 | 1.55±0.06 | -0.51 | -1.00 | -0.70 |
| 31 | Unnamed | Unknown | - | - | 0.12±0.02 | 0 | 0 | 0 |
| 32 | Unnamed | Unknown | - | - | 0.18±0.02 | 0 | 0 | 0 |
| | | Total | 68.23±2.42 | 62.12±2.72 | 53.78±2.48 | -0.34 | -0.47 | -0.39 |

* Omitted in WGI database

[Figure]

**Figure 2.** Changes in glacierized area of Kazbegi-Dzhimara massifs between 1960-1986-2014. See Tables 4 for the change statistics of individual glaciers. The 28/08/2014 Landsat 8 images was used as background.

[Figure]

**Figure 3.** Cumulative glacier area values for seven size classes in 1960-1986-2014 for the northern Greater Caucasus.

[Figure]

**Figure 4.** Cumulative glacier number values for seven size classes in 1960-1986-2014 for the northern Greater Caucasus.

[Figure]

**Figure 5.** Cumulative glacier area values for seven size classes in 1960-1986-2014 for the southern Greater Caucasus.

[Figure]

**Figure 6.** Cumulative glacier number values for seven size classes in 1960-1986-2014 for the southern Greater Caucasus.

[Figure]

**Figure 7.** Cumulative glacier area values for seven size classes in 1960-1986-2014 for western Greater Caucasus.

[Figure]

**Figure 8.** Cumulative glacier number values for seven size classes in 1960-1986-2014 for the western Greater Caucasus.

[Figure]

**Figure 9.** Cumulative glacier area values for seven size classes in 1960-1986-2014 for central Greater Caucasus.

[Figure]

**Figure 10.** Cumulative glacier number values for seven size classes in 1960-1986-2014 for the central Greater Caucasus.

[Figure]

**Figure 11.** Cumulative glacier area values for seven size classes in 1960-1986-2014 for eastern Greater Caucasus.

[Figure]

**Figure 12.** Cumulative glacier number values for seven size classes in 1960-1986-2014 for the eastern Greater Caucasus.

**Table 5.** Characteristics of glaciers used for measuring length change. The average error terms are ±15 m.

| Name/WGI ID | River basin | Area 1960 | Area 1986 | Area 2014 | 1960-1986 % yr[1] | 1986-2014 % yr[1] | 1960-2014 % yr[1] | Length change 1960-1986 | | Length change 1986-2014 | | Length change 1960-2014 | |
|---|---|---|---|---|---|---|---|---|---|---|---|---|---|
| | | | | | | | | m | m yr[1] | m | m yr[1] | m | m yr[1] |
| | | | | | | | | | | | | **Glaciers with >10 km² area** | |
| Bezingi | Cherek-Bezingskiy | 40.42±0.98 | 39.98±0.9 | 37.47±0.94 | -0.04 | -0.22 | -0.13 | -519 | -19.7 | -374 | -13.4 | -893 | -16.5 |
| Dych-sy-Ailama | Cherek-Balkarskiy | 39.49±0.98 | 34.85±0.94 | 27.53±0.78 | -0.45 | -0.75 | -0.56 | -461 | -17.7 | -1094 | -39.1 | -1555 | -28.8 |
| Karaugom | Karaugom | 29.94±0.6 | 29.17±0.62 | 23.99±0.44 | -0.09 | -0.63 | -0.36 | -164 | -6.3 | -1366 | -48.8 | -1530 | -28.3 |
| Lekhziri | Enguri | 35.80±0.9 | 33.95±0.94 | 23.76±0.72 | -0.19 | -1.07 | -0.62 | -859 | -33.0 | -736 | -26.3 | -1595 | -29.5 |
| Agashtan | Cherek-Balkarskiy | 21.35±0.36 | 20.39±0.32 | 18.93±0.44 | -0.17 | -0.25 | -0.20 | -368 | -14.2 | -587 | -21.0 | -955 | -17.7 |
| Midjirgi | Cherek-Bezingskiy | 13.77±0.5 | 13.90±0.48 | 12.71±0.48 | +0.03 | -0.30 | -0.14 | -808 | -31.1 | +40 | +1.4 | -768 | -14.2 |
| Tsaneri southern* | Enguri | 28.26±0.52 | 14.38±0.32 | 12.31±0.32 | -0.21 | -0.51 | -1.04 | -448 | -17.2 | -781 | -27.9 | -1229 | -22.8 |
| Tseya | Tseyadon | 14.03±0.42 | 12.83±0.38 | 11.87±0.36 | -0.32 | -0.26 | -0.28 | -295 | -11.3 | -341 | -12.2 | -636 | -11.8 |
| Tsaneri northern | Enguri | -** | 13.30±0.22 | 11.28±0.22 | -0.58 | -0.54 | - | - | - | -574 | -20.5 | - | - |
| | | | | | | | | | | | | **Glaciers with 5-10 km² area** | |
| Kvitlodi | Enguri | 12.23±0.26 | 11.65±0.24 | 9.58±0.2 | -0.18 | -0.63 | -0.40 | -598 | -23.0 | -883 | -31.5 | -1481 | -27.4 |
| Adishi | Enguri | 10.48±0.22 | 10.34±0.2 | 9.58±0.2 | -0.05 | -0.26 | -0.15 | -124 | -4.8 | -390 | -13.9 | -514 | -9.5 |
| Challaati | Enguri | 12.71±0.36 | 12.36±0.38 | 9.24±0.28 | -0.10 | -0.90 | -0.50 | -460 | -17.7 | -223 | -8.0 | -683 | -12.6 |
| Khalde | Enguri | 11.87±0.38 | 10.65±0.36 | 8.59±0.26 | -0.39 | -0.69 | -0.51 | -130 | -5.0 | -130 | -4.6 | -260 | -4.8 |
| Shkhelda | Baksan | 13.61±0.48 | 12.50±0.50 | 8.28±0.65 | -0.31 | -1.20 | -0.72 | -950 | -36.5 | -480 | -17.1 | -1430 | -26.5 |
| Bashil | Chegem | 8.16±0.19 | 7.91±0.19 | 7.34±0.19 | -0.11 | -0.25 | -0.18 | -230 | -8.8 | -530 | -18.9 | -760 | -14.1 |
| Dolra | Enguri | 7.95±0.21 | 6.44±0.16 | 5.36±0.12 | -0.73 | -0.59 | -0.60 | -595 | -22.9 | -160 | -5.7 | -755 | -14.0 |
| | | | | | | | | | | | | **Glaciers with 1-5 km² area** | |
| Boko | Rioni | 5.07±0.12 | 4.71±0.12 | 4.62±0.11 | -0.27 | -0.07 | -0.16 | -287 | -11.0 | -436 | -15.6 | -723 | -13.4 |
| Mostotsete | Urukh Headwaters | 4.27±0.14 | 3.58±0.012 | 3.23±0.13 | -0.62 | -0.34 | -0.45 | -105 | -4.0 | -135 | 4.8 | -240 | -4.4 |
| Marukh northern | Malii Zelenchuk | 3.25±0.08 | 3.30±0.08 | 2.82±0.07 | +0.05 | -0.51 | -0.24 | -255 | -9.8 | -240 | -8.6 | -495 | -9.2 |
| Chungurjar | Ullukam | 3.13±0.09 | 2.11±0.07 | 1.88±0.07 | -1.25 | -0.39 | -0.74 | -490 | -18.8 | -405 | -14.4 | -895 | -16.6 |
| Tbilisa | Riorni | 2.90±0.10 | 2.21±0.09 | 1.91±0.08 | -0.91 | -0.48 | -0.63 | -186 | 7.2 | -354 | -12.6 | -540 | -10.0 |
| Sakeni | Kodori | 2.47±0.07 | 2.39±0.08 | 1.99±0.05 | -0.12 | -0.59 | -0.35 | -560 | -21.5 | -275 | -9.8 | -835 | -15.5 |
| Abano | Tergi | 1.96±0.09 | 1.49±0.09 | 1.33±0.09 | -0.92 | -0.38 | -0.60 | -550 | -21.2 | -240 | -8.6 | -790 | -14.6 |
| | | | | | | | | | | | | **Glaciers with <1 km² area** | |

| SU5T09106388 | Rioni | 0.86±0.05 | 0.73±0.04 | 0.69±0.03 | -0.58 | -0.20 | -0.36 | -360 | -13.8 | -70 | -2.5 | -430 | -8.0 |
|---|---|---|---|---|---|---|---|---|---|---|---|---|---|
| *** | Sharo Argun | 0.90±0.05 | 0.77±0.04 | 0.55±0.03 | -0.55 | -1.02 | -0.72 | -65 | -2.5 | -270 | -9.6 | -335 | -6.2 |
| SU4G08011072 | Tergi | 0.62±0.03 | 0.55±0.03 | 0.42±0.02 | -0.43 | -0.84 | -0.60 | -60 | -2.3 | -310 | -11.1 | -370 | -6.9 |
| *** | Andiiskoe Koisu | 0.63±0.04 | 0.43±0.03 | 0.29±0.02 | -1.22 | -1.16 | -0.99 | -243 | -9.3 | -245 | -8.8 | -588 | -10.9 |
| SU4G08007139 | Cherek-Balkarskiy | 0.36±0.03 | 0.37±0.03 | 0.26±0.02 | +0.10 | -1.06 | -0.51 | -189 | -7.3 | -210 | -7.5 | -399 | -7.4 |
| SU4G08011083 | Tergi | 0.99±0.04 | 0.55±0.03 | 0.15±0.01 | -1.70 | -2.59 | -1.57 | -234 | -9.0 | -470 | -16.8 | -704 | -13.0 |
| SU5T09105282 | Enguri | 0.19±0.02 | 0.13±0.01 | 0.10±0.005 | -1.21 | -0.82 | -0.87 | -60 | -2.3 | -60 | -2.1 | -120 | -2.2 |

*, ** Until the 1980s the Southern and Northern Tsaneri were merged as one compound-valley type glacier. Their division likely happened in 1980-1985.

*** Omitted in WGI database.

[Figure]

**Figure 13.** Midjirgi Glacier advance between 1986-2000. (a) Landsat 5 TM, 6/08/1986. (b) Landsat 5 TM, 12/08/2000. In 1986, the flow of meltwater comes from a different position at the terminus. (c) With the two snout comparison, it is visible that the snout has advanced.

[Figure]

**Figure 14.** (a) Bezingi, (b) Karaugom, (c) Kvitlodi and (d) Adishi glaciers reduction in the years 1960-1986-2014. The 3/08/2014 Landsat 8 image (Table 1) is used as background.

---

## Referee Report (RR1)

**Re-review of the study by Tielidze and Wheate**

**General comments**

The revised paper by Tielidze and Wheate is rather different than the one before and in my opinion it has greatly improved. The authors have removed unnecessary contents and speculative statements and focused on the presentation of the results they have achieved. This makes the entire study more comprehensive and to the point. Apart from several minor points that I have listed in the specific comments, I have now only one major remaining issue, the limited science that is presented. With its current focus on data presentation the study would have been more appropriate for journals such as ESSD that do not require new exciting scientific progress. As this is maybe difficult to achieve at this stage I would like to make a few suggestions for adding some more science. I assume that most of the requested material has already been produced so I consider all these changes as minor.

A) Many of the tables presenting data of individual glaciers have been moved to the supplement. This is fine. On the other hand the paper itself contains aggregate figures (Fig. 4, 5, 6, 8, 10) of glacier statistics and changes that do not reveal any numbers. As these are also not listed in the supplement, the related values should be added in tabular form to the paper (Figs. 5 and 6d) and in the supplement (Figs. 4, 6abc, 8, 10).

B) The analysis is currently restricted to the presentation of area changes (Figs. 4 and 6) and some selected glacier statistics in the form of bar charts and plots (Figs. 5, 7 and 8). Overlay of glacier outlines is visualized in the supplement (Figs. 1, 2, 14). To put some 'meat to the bone' I would like to see some further plots and analysis of the data. Suggestions include (see also Specific comments):

a) Glacier aspect vs. mean elevation, maybe with colour-coded dots distinguishing between the northern and southern macro slope (as Fig. 8c)

b) A map showing the spatial distribution of mean elevation (using colour-coded circles) for all glaciers larger than a certain threshold (maybe $> 1$ km$^2$).

c) A map, plot or table showing the change in mid-point or mean elevation from the 1960s inventory to the most recent one. Plotting this against the change in minimum elevation could be very interesting as well.

d) A scatter plot showing glacier size vs. minimum and maximum elevation (dots in different colours and/or small symbols).

e) A scatter plot showing glacier size vs. relative change in area, maybe colour-coded for different regions and symbol coded for the two periods 1960-1986 and 1986-2014.

f) A scatter plot showing length changes vs. original length (this could end up in the supplement if results show a limited correlation).

C) The discussion section is currently looking at area change rates and differences to the RGI / GLIMS databases. This is fine but it should be extended, also considering the new plots suggested above. I thus suggest introducing three or four subheadings: 5.1 Glacier inventory parameters, 5.2 Glacier changes, 5.3 Comparison to GLIMS and the RGI, 5.4 Accuracy considerations (this can also be part of 5.2). The discussion should then focus on commenting on and evaluating the results rather than repeating the data. This might include interesting local differences, the large-scale variability and comparison to other studies, at best from the Alps that have at least similar characteristics (east-west and north-south gradients, similar mean elevation and glacier types).

Such an extended discussion would help to markedly shift the contents from a data report to a scientific paper with new insights. In this case it should clearly be in TC rather than ESSD. I hope the authors agree with this and can implement the suggested improvements.

**Specific comments**

P1

L19: aspect and height; what about slope?

L19-22: Simplify sentences, they have too many commas.

L23: The area results for the three inventories are fine but what about adding some further key characteristics and changes, e.g. mean size and elevation for the southern/northern slope, change in mean elevation from 1960 to 2014, and aspect dependence of elevation (if present)? See also General comments.

L24: 'The new glacier inventory will be …'

L30: Bliss et al. 2014 is maybe more a global scale hydrological study rather than a global sea-level study. Maybe move this one upward and add here Radic and Hock 2013 and/or Huss and Hock 2015?

L40: 'with ten thousands of people'?

P2

L1 I fully agree with you that this should be changed but it is just a matter of an email to Bruce Raup and than it is done. It is thus a rather short-lived statement and the reader might wonder why you have not already addressed this issue. It is much better to write here positively that the former wrong assignment of the greater Caucasus to one country has now been amended and split into three of them (R/G/A).

L6: This also sounds like complaining. In fact, nobody volunteered to write something so it is not covered (as many other regions). The GLIMS book never intended to be spatially complete as it lives from their contributors. So maybe remove this statement or write more precisely: 'As nobody volunteered to write a section about the Glaciers in the Caucasus for the GLIMS book (Kargel et al. 2014), the region is missing in this compilation.' or something similar.

L7/8: To avoid brackets, maybe write 'Our inventory has 6.5% less glacier area than … and 7.3% more than …'

L10: I suggest writing: 'Caucasus region based on manual delineation of glaciers from multi-temporal satellite images, and …'

P3

L6-9: This sounds a bit like mass balance observations are also providing temperature and precipitation time series. Please add one more sentence to clarify.

L14: 'photographs covering the time period 1875-1906 …'

L28: with a total area of 563.7 …

L37: maybe add: 'indicating a contrasting slow-down / increase of the loss rate on the northern/southern slopes.'

P4

L25: And for glaciers without a melt water stream or a very lateral location of it?

L33: interval of 20 m from

L36: 'georectified' you mean orthorectified (with a DEM) or geocoded?

P5

L1: not for deriving the mean slope? This is a very good indicator for mean ice thickness.

L5: Table formatting is maybe not required but I would align all heading row text centre, all cell body text (columns 2, 3, 5) left and cell body numbers right. As the path-row is visually difficult to extract from the given scene ID, I suggest adding a further column Path-Row (with entries only for Landsat and maybe ASTER).

L7: Actually both, the buffer method and the multiple digitizing only provide uncertainty (or precision), error (or accuracy) can only be determined by a comparison to appropriate reference data. So I suggest merging sections 3.2 and 3.3 to one section 'Uncertainty assessment' and start with a sentence saying that you have determined uncertainty with two independent methods. Than start with the buffer method (as it is the more simple one) before you describe the multiple digitizing.

When you also want to include the accuracy assessment performed with the Garmin GPS, name this section 'Uncertainty and accuracy assessment'. Then introduce the latter and also present results of it as these are currently missing. I see black dots in Fig. 2 c/d but it is not described in the caption what they mean (please add). Regarding Fig. 2, please make the a b c d panel marks much larger (factor 4) than they are now. Also be consistent with the syntax: Either use (a) (b) as in Figs. 1 and 8 or just a b as here.

L17/18: Maybe write: 'To determine the precision of the digitizing, we manually digitized fifteen differently sized glaciers independently five times in the western …'? (it is not really the error that is determined by the method but the variability of the interpretation. So roughly this is the analysts precision.

L28/9: 'covered heavily by debris'

P6

L6: It is difficult to see anything on Fig 2c. I suggest replacing it with another close-up view.

L9: As mentioned above, please merge this section with 3.2.

L16/17: You might consider to also referring here to your own results shown in Figs. 2a and b. They clearly reveal a ±1/2 pixel buffer for clean ice and a ±1 or 2 pixel buffer for debris-covered ice.

L20: 30 m (with a space in-between)

P7

L6: I suggest showing a close-up of the debris-covered part.

P8

Tables 2 and 3: I think these two tables can be safely merged.

Figure 4: I suggest adding minor tick marks on the y-axis (step 0.1) and repeat them on the opposite site

L15: experienced the highest relative glacier area loss

L17/18: This might be correct but it comes a bit of a sudden and is difficult to verify without knowing further details. So maybe move it to the discussion and add some details about the cited study there. The problem is that you cite here a study that has been published before this study but you link the results of this study to it. So I wonder how the cited study could have known the results presented here?

L21: I would not introduce a subheading 4.1.1 when there is no 4.1.2. Maybe rename 4.1 to 'Glacier changes for the entire study region' and 4.1.2 to '4.2 Glacier changes in the … massif?

P9

L24 (Fig. 5): I think this figure is fine in general but I would change a few things: remove the 'Mean area …' text line, add some minor tick marks for both y-axes (left and right), use more distinct colours (or shades of grey?) for the bars and triangles (green is difficult to distinguish from cyan), and in particular change the colour of the 1960 triangle to something else (black?). As it is very difficult to extract any numbers from the graph, please also provide a table listing all numbers (can be in an Appendix).

L30: resulting from the disintegration of … (retreat is change in terminus position).

L34: As Fig. 6 is only an aggregate figure, would it be possible to add a scatter plot showing the individual values, maybe symbol coded for the two periods (1960-1986 and 1986-2014) and colour coded for northern / southern slopes? This would also depict the local variability.

P10

L2: Please add the equidistance of the elevation bands used. Currently it looks like 250 m? In this case please use 50 m to avoid the blocky appearance of the graph.

L9 (Fig. 7): Can you please add some minor tick marks on both axes. With 50 m elevation bins please also use major gridlines for the y-axis.

L9: Please consider adding a scatter plot (as Fig. 7b) showing glacier size vs. minimum and maximum elevation (colour coded in the same plot).

P11

L3 (Fig. 8): Please use capital letters for the cardinal directions (N, NE, E, etc.)

L3: Please add a scatter plot (as Fig. 8b) showing aspect vs. mean or mid-point elevation of glaciers

L3: If there is some interesting variability, please show a map with colour-coded circles representing glacier mean (or mid-point) elevation for all glaciers larger 0.5 or 1 km$^2$.

L5: When analysing length changes, wouldn't it be more sensible to sort glaciers for length classes? I ask because there is often a certain relationship between initial glacier length and length change. You can check this by creating a scatter plot initial length vs. (absolute) length change. If there is a relation, I suggest showing also this plot.

L6/7: 'length change': I think you mean 'retreat rates' here?

L28: Discussion section: The two topics presented in the discussion section are fine. However, I think ones the additional plots are shown some further discussion on the achieved results should be presented. This should also add some science (or give some 'meat to the bone') to this data-driven contribution.

L36: than from 1957-2000

L37ff: Interpretation of shrinkage rates: as mentioned in my previous review, I would really restrict the higher loss rates in regions with smaller glaciers to their size and nothing else (please add the related scatterplot as suggested above). All other representations require knowledge about changes that you do not have. I repeat: Glaciers are where they are because climate is at it is. So climatic conditions itself do not have any impact on area change rates. What you need to show for your statements is that climate CHANGE was different in different regions and elevations. For the eastern Caucasus this means that climate has dried (less precipitation) at the elevation of glaciers whereas at the same time the larger accumulation areas of the glaciers in the central Caucasus have received more precipitation. Similarly, to get increased glacier loss at lower elevations you must show that temperatures have increased more at these elevations than higher up. As no proof is given for any of these trends, you cannot claim that these are the reasons.

P12

L21/2: I assume these differences have a sign? Can you please add which ones are larger and smaller?

L26 (Fig. 10): I think there is not much to see when absolute area differences are plotted like this. Better use a plot style like in Fig, 6 with bars and positive/negative differences. Maybe these display even better when presented as relative rather than absolute differences.

P13

L8/9: Is this shown somewhere (map overestimation of snowfields and the related correction with Corona images? I suggest adding this, as it would have some relevance beyond this study. Increasing the consistency in the interpretation of glacier outlines is still a major issue for glacier inventories so a practical example would be very helpful.

L20/1: 0.7% is quite strong but it is only half of the rate in the Alps. For this not further elaborated statement I would maybe add a citation from one of the global scale studies presenting volume changes per RGI region until 2100.

---

## Author Response (AR2)

**Response letter for reviewer and editor comments**

**"The Greater Caucasus Glacier Inventory (Russia/Georgia/Azerbaijan)" by L. G. Tielidze and R. D. Wheate**

http://www.the-cryosphere-discuss.net/tc-2017-48/
doi.org/10.5194/tc-2017-48

Dear referee and editor,

We very much appreciate your comments concerning our manuscript. Those comments are valuable and helpful for improving our manuscript. We tried our best to follow all comments and made revision and answers carefully. All corrections are marked in yellow in the revised manuscript.
Please see detail answers on the comments below.

Thank you
* * *
*A) Many of the tables presenting data of individual glaciers have been moved to the supplement. This is fine. On the other hand the paper itself contains aggregate figures (Fig. 4, 5, 6, 8, 10) of glacier statistics and changes that do not reveal any numbers. As these are also not listed in the supplement, the related values should be added in tabular form to the paper (Figs.5 and 6d) and in the supplement (Figs. 4, 6abc, 8, 10).*

We have changed and added some figures/tables: Fig. 4; Table 3 in the manuscript and Table 5-6; Fig. 3-12 in the supplement.

*B) The analysis is currently restricted to the presentation of area changes (Figs. 4 and 6) and some selected glacier statistics in the form of bar charts and plots (Figs. 5, 7 and 8). Overlay of glacier outlines is visualized in the supplement (Figs. 1, 2, 14). To put some 'meat to the bone' I would like to see some further plots and analysis of the data. Suggestions include (see also Specific comments):*

*a) Glacier aspect vs. mean elevation, maybe with colour-coded dots distinguishing be- tween the northern and southern macro slope (as Fig. 8c)*

*b) A map showing the spatial distribution of mean elevation (using colour-coded circles) for all glaciers larger than a certain threshold (maybe > 1 $km^2$).*

*c) A map, plot or table showing the change in mid-point or mean elevation from the 1960s inventory to the most recent one. Plotting this against the change in minimum elevation could be very interesting as well.*

*d) A scatter plot showing glacier size vs. minimum and maximum elevation (dots in different colours and/or small symbols).*

*e) A scatter plot showing glacier size vs. relative change in area, maybe colour-coded for different regions and symbol coded for the two periods 1960-1986 and 1986-2014.*

*f) A scatter plot showing length changes vs. original length (this could end up in the supplement if results show a limited correlation).*

Please see new figures/tables: Fig. 8c; Table 3; Fig. 7b.

*C) The discussion section is currently looking at area change rates and differences to the RGI/GLIMS databases. This is fine but it should be extended, also considering the new plots suggested above. I thus suggest introducing three or four subheadings: 5.1 Glacier inventory parameters, 5.2 Glacier changes, 5.3 Comparison to GLIMS and the RGI, 5.4 Accuracy con-*

*siderations (this can also be part of 5.2). The discussion should then focus on commenting on and evaluating the results rather than repeating the data. This might include interesting local differences, the large-scale variability and comparison to other studies, at best from the Alps that have at least similar characteristics (east-west and north-south gradients, similar mean elevation and glacier types).*

We agree. Please see new Discussion section P 12-13.

*Specific comments*
*P1*
*L19: aspect and height; what about slope?*

We add slope data P1 L19; P9 Table 3; P12 Fig. 9.

*L19-22: Simplify sentences, they have too many commas.*

We have changed this, removing some commas.

*L23: The area results for the three inventories are fine but what about adding some further key characteristics and changes, e.g. mean size and elevation for the southern/northern slope, change in mean elevation from 1960 to 2014, and aspect dependence of elevation (if present)? See also General comments.*

We agree, please see P9 Table 3.

*L24: 'The new glacier inventory will be …'*

We have changed this sentence P1 L25.

*L30: Bliss et al. 2014 is maybe more a global scale hydrological study rather than a global sea-level study. Maybe move this one upward and add here Radic and Hock 2013 and/or Huss and Hock 2015?*

We have added new reference P1 L32.

*L40: 'with ten thousands of people'?*

We have changed this P1 L43.

*P2*
*L1 I fully agree with you that this should be changed but it is just a matter of an email to Bruce Raup and than it is done. It is thus a rather short-lived statement and the reader might wonder why you have not already addressed this issue. It is much better to write here positively that the former wrong assignment of the greater Caucasus to one country has now been amended and split into three of them (R/G/A).*

We have changed this sentence P1 L25-27.

*L6: This also sounds like complaining. In fact, nobody volunteered to write something so it is not covered (as many other regions). The GLIMS book never intended to be spatially complete as it lives from their contributors. So maybe remove this statement or write more precisely: 'As no- body volunteered to write a section about the Glaciers in the Caucasus for the GLIMS book (Kargel et al. 2014), the region is missing in this compilation.' or something similar.*

We have changed this sentence P2 L6-8.

*L7/8: To avoid brackets, maybe write 'Our inventory has 6.5% less glacier area than … and 7.3% more than …'*

We have changed this sentence P2 L5-6.

*L10: I suggest writing: 'Caucasus region based on manual delineation of glaciers from multi-temporal satellite images, and …'*

We agree P2 L10.

*P3*
*L6-9: This sounds a bit like mass balance observations are also providing temperature and precipita- tion time series. Please add one more sentence to clarify.*

We decided to delete this sentence as it was more connected mass balance and less glacier inventory.

*L14: 'photographs covering the time period 1875-1906 …'*

We agree P3 L12.

*L28: with a total area of 563.7 …*

| |
|---|
| We agree P3 L27. |
| *L37: maybe add: 'indicating a contrasting slow-down/increase of the loss rate on the north-ern/southern slopes.'* |
| We agree P3 L37. |
| *P4*
 *L25: And for glaciers without a melt water stream or a very lateral location of it?* |
| We have changed this sentence P4 L24-25. |
| *L33: interval of 20 m from* |
| We agree P4 L33. |
| *L36: 'georectified' you mean orthorectified (with a DEM) or geocoded?* |
| Georectified is the correct word as we didn't use a DEM. |
| *P5*
 *L1: not for deriving the mean slope? This is a very good indicator for mean ice thickness.* |
| We have add slope data P9 Table 3; P12 Fig. 9. |
| *L5: Table formatting is maybe not required but I would align all heading row text centre, all cell body text (columns 2, 3, 5) left and cell body numbers right. As the path-row is visually difficult to ex- tract from the given scene ID, I suggest adding a further column Path-Row (with entries only for Landsat and maybe ASTER).* |
| We agree and changed table P5 Table 5. |
| *L7: Actually both, the buffer method and the multiple digitizing only provide uncertainty (or precision), error (or accuracy) can only be determined by a comparison to appropriate reference data. So I suggest merging sections 3.2 and 3.3 to one section 'Uncertainty assessment' and start with a sentence saying that you have determined uncertainty with two independent methods. Than start with the buffer method (as it is the more simple one) before you describe the multiple digitizing. When you also want to include the accuracy assessment performed with the Garmin GPS, name this section 'Uncertainty and accuracy assessment'. Then introduce the latter and also present results of it as these are currently missing. I see black dots in Fig. 2 c/d but it is not described in the caption what they mean (please add). Regarding Fig. 2, please make the a b c d panel marks much larger (factor 4) than they are now. Also be consistent with the syntax: Either use (a) (b) as in Figs. 1 and 8 or just a b as here.* |
| We agree, please see the new "3.2 Glacier uncertainty and accuracy assessment" section P5-7. |
| *L17/18: Maybe write: 'To determine the precision of the digitizing, we manually digitized fifteen differently sized glaciers independently five times in the western …'? (it is not really the error that is determined by the method but the variability of the interpretation. So roughly this is the analysts precision.* |
| We have changed this sentence P6 L6-7 |
| *L28/9: 'covered heavily by debris'* |
| We changed this sentence P6 L25 |
| *P6*
 *L6: It is difficult to see anything on Fig 2c. I suggest replacing it with another close-up view.* |
| We have changed this figure, please see P6 Figure 2. |
| *L9: As mentioned above, please merge this section with 3.2.* |
| We agree, please see the new "3.2 Glacier uncertainty and accuracy assessment" section P5-7. |
| *L16/17: You might consider to also referring here to your own results shown in Figs. 2a and b. They clearly reveal a ±1/2 pixel buffer for clean ice and a ±1 or 2 pixel buffer for debris-covered ice.* |
| We have added our results here P5 L20. |
| *L20: 30 m (with a space in-between)* |
| We have changed this P5 L20. |
| *P7*
 *L6: I suggest showing a close-up of the debris-covered part.* |
| We agree, please see P7 Figure 3d. |

| | |
|---|---|
| *P8*
 *Tables 2 and 3: I think these two tables can be safely merged.* | |
| We agree, please see P8 Table 2. | |
| *Figure 4: I suggest adding minor tick marks on the y-axis (step 0.1) and repeat them on the opposite site* | |
| We have changed this figure, please see P8 Figure 4. | |
| *L15: experienced the highest relative glacier area loss* | |
| We agree, please see P8 L20. | |
| *L17/18: This might be correct but it comes a bit of a sudden and is difficult to verify without knowing further details. So maybe move it to the discussion and add some details about the cited study there. The problem is that you cite here a study that has been published before this study but you link the results of this study to it. So I wonder how the cited study could have known the results presented here?* | |
| We have decided to delete this sentence, as it was a bit confusing. | |
| *L21: I would not introduce a subheading 4.1.1 when there is no 4.1.2. Maybe rename 4.1 to 'Glacier changes for the entire study region' and 4.1.2 to '4.2 Glacier changes in the … massif?* | |
| We agree, please see P7 L11; P8 L27. | |
| *P9*
 *L24 (Fig. 5): I think this figure is fine in general but I would change a few things: remove the 'Mean area …' text line, add some minor tick marks for both y-axes (left and right), use more distinct colours (or shades of grey?) for the bars and triangles (green is difficult to distinguish from cyan), and in particular change the colour of the 1960 triangle to something else (black?). As it is very difficult to extract any numbers from the graph, please also provide a table listing all numbers (can be in an Appendix).* | |
| We agree, please see new figure P9 Figure 5. | |
| *L30: resulting from the disintegration of … (retreat is change in terminus position).* | |
| We have changed this P10 L6. | |
| *L34: As Fig. 6 is only an aggregate figure, would it be possible to add a scatter plot showing the individual values, maybe symbol coded for the two periods (1960-1986 and 1986-2014) and colour coded for northern/southern slopes? This would also depict the local variability.* | |
| This is an insightful suggestion, but we were reluctant to add another 'similar' figure to those already in the paper. The north-south values are already shown in figures 4,7-9 and table 3. | |
| *P10*
 *L2: Please add the equidistance of the elevation bands used. Currently it looks like 250 m? In this case please use 50 m to avoid the blocky appearance of the graph.* | |
| We agree and changed figure, please see P11 Figure 7a. | |
| *L9 (Fig. 7): Can you please add some minor tick marks on both axes. With 50 m elevation bins please also use major gridlines for the y-axis.* | |
| We have changed this figure according your comment, please see new figure P11 Figure 7a. | |
| *L9: Please consider adding a scatter plot (as Fig. 7b) showing glacier size vs. minimum and maximum elevation (colour coded in the same plot)* | |
| We agree and have added new figure, please see P11 Figure 7b. | |
| *P11*
 *L3 (Fig. 8): Please use capital letters for the cardinal directions (N, NE, E, etc.)* | |
| We agree and changed these figures, please see P11 Figure 8a-c | |
| *L3: Please add a scatter plot (as Fig. 8b) showing aspect vs. mean or mid-point elevation of glaciers* | |
| We have added new figure, please see P11 Figure 8c. | |
| *L3: If there is some interesting variability, please show a map with colour-coded circles representing glacier mean (or mid-point) elevation for all glaciers larger 0.5 or 1 km$^2$.* | |
| We have added new table for glacier mean and minimum elevation P9 Table3 | |

| |
|---|
| *L5: When analysing length changes, wouldn't it be more sensible to sort glaciers for length classes? I ask because there is often a certain relationship between initial glacier length and length change. You can check this by creating a scatter plot initial length vs. (absolute) length change. If there is a relation, I suggest showing also this plot.* |
| We reviewed this, but did not find a significant relationship between glacier length and change worth showing in this paper. |
| *L6/7: 'length change': I think you mean 'retreat rates' here?* |
| We have changed this P12 L6. |
| *L28: Discussion section: The two topics presented in the discussion section are fine. However, I think ones the additional plots are shown some further discussion on the achieved results should be presented. This should also add some science (or give some 'meat to the bone') to this data-driven contribution.* |
| We agree, please see new Discussion section P12-14. |
| *L36: than from 1957-2000* |
| We have changed this P13 L1. |
| *L37ff: Interpretation of shrinkage rates: as mentioned in my previous review, I would really restrict the higher loss rates in regions with smaller glaciers to their size and nothing else (please add the related scatterplot as suggested above). All other representations require knowledge about changes that you do not have. I repeat: Glaciers are where they are because climate is at it is. So climatic conditions itself do not have any impact on area change rates. What you need to show for your statements is that climate CHANGE was different in different regions and elevations. For the eastern Caucasus this means that climate has dried (less precipitation) at the elevation of glaciers whereas at the same time the larger accumulation areas of the glaciers in the central Caucasus have received more precipitation. Similarly, to get increased glacier loss at lower elevations you must show that temperatures have increased more at these elevations than higher up. As no proof is given for any of these trends, you cannot claim that these are the reasons.* |
| We decided to delete this sentence and added new text P13 L1-5. |
| *P12* *L21/2: I assume these differences have a sign? Can you please add which ones are larger and smaller?* |
| We have added exact numbers P14 L12-13. |
| *L26 (Fig. 10): I think there is not much to see when absolute area differences are plotted like this. Better use a plot style like in Fig, 6 with bars and positive/negative differences. Maybe these display even better when presented as relative rather than absolute differences* |
| We have changed this figure, please see P14 Figure 11 |
| *P13* *L8/9: Is this shown somewhere (map overestimation of snowfields and the related correction with Corona images? I suggest adding this, as it would have some relevance beyond this study. Increasing the consistency in the interpretation of glacier outlines is still a major issue for glacier inventories so a practical example would be very helpful.* |
| We have added new figure for this, please see P6 Figure 2d-e |
| *L20/1: 0.7% is quite strong but it is only half of the rate in the Alps. For this not further elaborated statement I would maybe add a citation from one of the global scale studies presenting volume changes per RGI region until 2100.* |
| We have added to the text: Huss and Hock, please see P15 L7-8. |

**The Greater Caucasus Glacier Inventory (Russia/Georgia/Azerbaijan)**

Levan G. Tielidze[1,3], Roger D. Wheate[2]

[1]Department of Geomorphology, Vakhushti Bagrationi Institute of Geography, Ivane Javakhishvili Tbilisi State University, 6 Tamarashvili st., Tbilisi, Georgia, 0177

[2]Natural Resources and Environmental Studies, University of Northern British Columbia, 3333 University Way, Prince George, BC, Canada, V2N 4Z9

[3]Department of Earth Sciences, Georgian National Academy of Sciences, 52 Rustaveli Ave., Tbilisi, Georgia, 0108

*Correspondence to:* Levan G. Tielidze (levan.tielidze@tsu.ge)

**Abstract**

There have been numerous studies of glaciers in the Greater Caucasus, but none that have generated a modern glacier database across the whole range. Here, we present an updated and expanded glacier inventory over the three 1960-1986-2014 time periods covering the entire Greater Caucasus. Large scale topographic maps and satellite imagery (Corona, Landsat 5, Landsat 8 and ASTER) were used to conduct a remote sensing survey of glacier change and the 30 m resolution ASTER GDEM to determine the aspect, slope and height distribution of glaciers. Glacier margins were mapped manually and reveal that in 1960, the mountains contained 2349 glaciers with a total glacier surface area of 1674.9±70.4 km$^2$. By 1986, glacier surface area had decreased to 1482.1±64.4 km$^2$ (2209 glaciers), and by 2014 to 1193.2±54.0 km$^2$ (2020 glaciers). This represents a 28.8±4.4% (481±21.2 km$^2$) or 0.53% yr$^{-1}$ reduction in total glacier surface area between 1960 and 2014 and an increase in the rate of area loss since 1986 (0.69% yr$^{-1}$), compared to 1960-1986 (0.44% yr$^{-1}$). Glacier mean size decreased from 0.70-0.66-0.57 km$^2$ between 1960-1986-2014. This new glacier inventory data have been submitted to GLIMS, and can be used as a basis dataset for future studies. The GLIMS former assignment of the greater Caucasus to one country has now been amended and split into three (Russian-Georgia-Azerbaijan).

[revised manuscript text omitted]

* Omitted in WGI database

[Figure]

**Figure 1.** Changes in glacierized area of Elbrus between 1960-1986-2014.
See Table 3 for the change statistics of individual glaciers. The 03/08/2014 Landsat 8 image is used as background.

**Table 4.** Kazbegi-Dzhimara massif glacier number and area change in 1960-1986-2014. All glaciers are shown in Fig. 2.

| | Kazbegi-Dzhimara massif glaciers | | Topographic maps 1960 | Landsat 5, 06/08/1986 | Landsat 8, 28/08/14/ | Area change | | |
|---|---|---|---|---|---|---|---|---|
| # | Name | WGI ID | Area km$^2$ | Area km$^2$ | Area km$^2$ | 1960-1986 % yr$^{-1}$ | 1986-2014 % yr$^{-1}$ | 1960-2014 % yr$^{-1}$ |
| 1 | Mydagrabyn | SU4G08010031 | 9.98±0.22 | 9.73±0.24 | 8.16±0.24 | -0.09 | -0.57 | -0.33 |
| 2 | Unnamed* | Unknown | - | - | 0.16±0.02 | 0 | 0 | 0 |
| 3 | Kolka | SU4G08010039 | 5.06±0.7 | 4.28±0.48 | 2.51±0.44 | -0.59 | -1.47 | -0.91 |
| 4 | Unnamed | Unknown | - | - | 0.75±0.04 | 0 | 0 | 0 |
| 5 | Unnamed | SU4G08010040 | 0.79±0.06 | 0.73±0.04 | 0.50±0.04 | -0.29 | -1.12 | -0.68 |
| 6 | Maili | SU4G08010041 | 7.29±0.14 | 6.75±0.14 | 6.57±0.12 | -0.28 | -0.09 | -0.18 |
| 7 | Unnamed | Unknown | - | - | 0.05±0.008 | 0 | 0 | 0 |
| 8 | Unnamed | SU4G08010042 | - | 0.67±0.04 | 0.44±0.02 | 0 | -1.22 | 0 |
| 9 | Chachi | SU4G08011046 | 2.61±0.06 | 2.57±0.08 | 1.85±0.08 | -0.05 | -1.00 | -0.53 |
| 10 | Unnamed | Unknown | - | - | 0.08±0.012 | 0 | 0 | 0 |
| 11 | Unnamed | SU4G08011047 | 0.86±0.06 | 0.52±0.04 | 0.12±0.02 | -1.52 | -2.74 | -1.59 |
| 12 | Unnamed | Unknown | - | - | 0.05±0.008 | 0 | 0 | 0 |
| 13 | Devdoraki | SU4G08011048 | 7.19±0.2 | 6.96±0.2 | 4.40±0.12 | -0.12 | -1.31 | -0.71 |
| 14 | Unnamed | Unknown | - | - | 1.78±0.08 | 0 | 0 | 0 |
| 15 | Abano | SU4G08011049 | 1.96±0.08 | 1.49±0.08 | 1.33±0.1 | -0.92 | -0.38 | -0.60 |
| 16 | Unnamed | Unknown | - | - | 0.03±0.004 | 0 | 0 | 0 |
| 17 | Unnamed | Unknown | 0.58±0.04 | 0.34±0.04 | 0.1±0.004 | -1.59 | -2.52 | -1.53 |
| 18 | Gergeti | SU4G08011052 | 6.82±0.18 | 6.26±0.2 | 5.77±0.18 | -0.31 | -0.27 | -0.28 |
| 19 | None | SU4G08011056 | 0.49±0.04 | 0.39±0.02 | 0.36±0.02 | -0.78 | -0.27 | -0.49 |
| 20 | Denkara | SU4G08011057 | 1.33±0.04 | 0.93±0.04 | 0.31±0.02 | -1.15 | -2.38 | -1.42 |
| 21 | Unnamed | Unknown | 0.49±0.04 | 0.06±0.01 | 0.03±0.004 | -3.37 | -1.78 | -1.73 |
| 22 | Unnamed | SU4G08011058 | 0.89±0.06 | 0.63±0.04 | 0.53±0.04 | -1.12 | -0.56 | -0.74 |
| 23 | Unnamed | SU4G08011059 | 1.12±0.04 | 0.98±0.06 | 0.75±0.06 | -0.48 | -0.83 | -0.61 |
| 24 | Mna | SU4G08011060 | 3.25±0.1 | 2.89±0.12 | 2.59±0.12 | -0.42 | -0.37 | -0.37 |
| 25 | Unnamed | SU4G08011061 | 1.57±0.04 | 1.30±0.06 | 1.44±0.04 | -0.66 | +0.38 | -0.15 |
| 26 | Suatisi Eastern | SU4G08011062 | 10.84±0.2 | 9.87±0.24 | 8.89±0.18 | -0.34 | -0.35 | -0.33 |

| 27 | Unnamed | Unknown | - | - | 0.08±0.012 | 0 | 0 | 0 |
| 28 | Suatisi Central | SU4G08011063 | 2.62±0.1 | 2.32±0.08 | 2.07±0.08 | -0.44 | -0.38 | -0.39 |
| 29 | Unnamed | Unknown | - | 0.29±0.04 | 0.23±0.02 | 0 | -0.73 | 0 |
| 30 | Suatisi Western | SU4G08011064 | 2.49±0.08 | 2.16±0.08 | 1.55±0.06 | -0.51 | -1.00 | -0.70 |
| 31 | Unnamed | Unknown | - | - | 0.12±0.02 | 0 | 0 | 0 |
| 32 | Unnamed | Unknown | - | - | 0.18±0.02 | 0 | 0 | 0 |
| | | **Total** | **68.23±2.42** | **62.12±2.72** | **53.78±2.48** | **-0.34** | **-0.47** | **-0.39** |

\* Omitted in WGI database

[Figure]

**Figure 2.** Changes in glacierized area of Kazbegi-Dzhimara massifs between 1960-1986-2014.
See Table 4 for the change statistics of individual glaciers. The 28/08/2014 Landsat 8 image is used as background.

**Table 5.** Cumulative glacier area and number change for seven size classes in the Greater Caucasus in 1960-1986-2014.

| Size class (km²) | Area | | | Number | | |
|---|---|---|---|---|---|---|
| | **1960** | **1986** | **2014** | **1960** | **1986** | **2014** |
| 0.01-0.05 | 14.95 | 12.26 | 16.21 | 431 | 364 | 516 |
| 0.05-0.1 | 33.70 | 38 | 28.43 | 427 | 502 | 388 |
| 0.1-0.5 | 219.18 | 194.42 | 154.09 | 918 | 839 | 695 |
| 0.5-1.0 | 173.27 | 153.38 | 127.00 | 241 | 209 | 175 |
| 1.0-5.0 | 555.28 | 514.46 | 415.29 | 275 | 246 | 204 |
| 5.0-10.0 | 242.12 | 208.08 | 210.29 | 35 | 29 | 29 |
| >10.0 | 436.28 | 361.68 | 241.72 | 22 | 20 | 13 |

[Figure]

**Figure 3.** Cumulative glacier area values for seven size classes in 1960-1986-2014 for the northern Greater Caucasus.

[Figure]

**Figure 4.** Cumulative glacier number values for seven size classes in 1960-1986-2014 for the northern Greater Caucasus.

[Figure]

**Figure 5.** Cumulative glacier area values for seven size classes in 1960-1986-2014 for the southern Greater Caucasus.

[Figure]

**Figure 6.** Cumulative glacier number values for seven size classes in 1960-1986-2014 for the southern Greater Caucasus.

[Figure]

**Figure 7.** Cumulative glacier area values for seven size classes in 1960-1986-2014 for western Greater Caucasus.

[Figure]

**Figure 8.** Cumulative glacier number values for seven size classes in 1960-1986-2014 for the western Greater Caucasus.

[Figure]

**Figure 9.** Cumulative glacier area values for seven size classes in 1960-1986-2014 for central Greater Caucasus.

[Figure]

**Figure 10.** Cumulative glacier number values for seven size classes in 1960-1986-2014 for the central Greater Caucasus.

[Figure]

**Figure 11.** Cumulative glacier area values for seven size classes in 1960-1986-2014 for eastern Greater Caucasus.

[Figure]

**Figure 12.** Cumulative glacier number values for seven size classes in 1960-1986-2014 for the eastern Greater Caucasus.

**Table 6.** y% Area change for the seven size classes glacier in the western, central, eastern sections and entire Greater Caucasus in 1960-1986-2014.

| Size class (km²) | Western Caucasus | | | Central Caucasus | | | Eastern Caucasus | | | Entire Greater Caucasus | | |
|---|---|---|---|---|---|---|---|---|---|---|---|---|
| | 1960-1986 y% | 1986-2014 y% | 1960-2014 y% | 1960-1986 y% | 1986-2014 y% | 1960-2014 y% | 1960-1986 y% | 1986-2014 y% | 1960-2014 y% | 1960-1986 y% | 1986-2014 y% | 1960-2014 y% |
| 0.01-0.05 | 0.42 | 1.47 | 1.05 | -0.96 | 1.42 | 0.08 | 1.05 | -0.43 | 0.21 | -0.69 | 1.15 | 0.15 |
| 0.05-0.1 | 0.69 | -0.1 | 0.26 | 0.85 | -0.85 | -0.12 | -0.88 | -1.39 | -0.98 | 0.49 | -0.89 | -0.28 |
| 0.1-0.5 | -0.77 | -0.6 | -0.37 | -0.28 | -0.46 | -0.36 | -1.07 | 1.8 | -1.18 | -0.43 | -0.74 | -0.55 |
| 0.5-1.0 | 0.09 | -0.93 | -0.44 | -0.14 | -0.07 | -0.1 | -1.67 | -1.61 | -1.27 | -0.44 | -0.61 | -0.49 |
| 1.0-5.0 | -0.65 | -0.78 | -0.65 | -0.008 | -0.6 | -0.31 | -0.83 | -0.98 | -0.8 | -0.28 | -0.68 | -0.46 |
| 5.0-10.0 | -0.59 | -1.16 | -0.8 | -0.74 | 0.43 | -0.17 | 2.45 | -2.43 | -0.88 | -0.54 | 0.03 | -0.24 |
| >10.0 | - | - | - | -0.57 | -1.18 | -0.8 | - | - | - | -0.65 | -1.18 | -0.82 |

**Table 7.** Characteristics of glaciers used for measuring length change. The average error terms are ±15 m.

| Name/WGI ID | River basin | Area 1960 | Area 1986 | Area 2014 | 1960-1986 % yr[1] | 1986-2014 % yr[1] | 1960-2014 % yr[1] | Length change 1960-1986 | | Length change 1986-2014 | | Length change 1960-2014 | |
|---|---|---|---|---|---|---|---|---|---|---|---|---|---|
| | | | | | | | | m | m yr[1] | m | m yr[1] | m | m yr[1] |
| | | | | | | | | | | | | **Glaciers with >10 km² area** | |
| Bezingi | Cherek-Bezingskiy | 40.42±0.98 | 39.98±0.9 | 37.47±0.94 | -0.04 | -0.22 | -0.13 | -519 | -19.7 | -374 | -13.4 | -893 | -16.5 |
| Dych-sy-Ailama | Cherek-Balkarskiy | 39.49±0.98 | 34.85±0.94 | 27.53±0.78 | -0.45 | -0.75 | -0.56 | -461 | -17.7 | -1094 | -39.1 | -1555 | -28.8 |
| Karaugom | Karaugom | 29.94±0.6 | 29.17±0.62 | 23.99±0.44 | -0.09 | -0.63 | -0.36 | -164 | -6.3 | -1366 | -48.8 | -1530 | -28.3 |
| Lekhziri | Enguri | 35.80±0.9 | 33.95±0.94 | 23.76±0.72 | -0.19 | -1.07 | -0.62 | -859 | -33.0 | -736 | -26.3 | -1595 | -29.5 |
| Agashtan | Cherek-Balkarskiy | 21.35±0.36 | 20.39±0.32 | 18.93±0.44 | -0.17 | -0.25 | -0.20 | -368 | -14.2 | -587 | -21.0 | -955 | -17.7 |
| Midjirgi | Cherek-Bezingskiy | 13.77±0.5 | 13.90±0.48 | 12.71±0.48 | +0.03 | -0.30 | -0.14 | -808 | -31.1 | +40 | +1.4 | -768 | -14.2 |
| Tsaneri southern* | Enguri | 28.26±0.52 | 14.38±0.32 | 12.31±0.32 | -0.21 | -0.51 | -1.04 | -448 | -17.2 | -781 | -27.9 | -1229 | -22.8 |
| Tseya | Tseyadon | 14.03±0.42 | 12.83±0.38 | 11.87±0.36 | -0.32 | -0.26 | -0.28 | -295 | -11.3 | -341 | -12.2 | -636 | -11.8 |
| Tsaneri northern | Enguri | -** | 13.30±0.22 | 11.28±0.22 | -0.58 | -0.54 | - | - | - | -574 | -20.5 | - | - |
| | | | | | | | | | | | | **Glaciers with 5-10 km² area** | |
| Kvitlodi | Enguri | 12.23±0.26 | 11.65±0.24 | 9.58±0.2 | -0.18 | -0.63 | -0.40 | -598 | -23.0 | -883 | -31.5 | -1481 | -27.4 |
| Adishi | Enguri | 10.48±0.22 | 10.34±0.2 | 9.58±0.2 | -0.05 | -0.26 | -0.15 | -124 | -4.8 | -390 | -13.9 | -514 | -9.5 |
| Challaati | Enguri | 12.71±0.36 | 12.36±0.38 | 9.24±0.28 | -0.10 | -0.90 | -0.50 | -460 | -17.7 | -223 | -8.0 | -683 | -12.6 |
| Khalde | Enguri | 11.87±0.38 | 10.65±0.36 | 8.59±0.26 | -0.39 | -0.69 | -0.51 | -130 | -5.0 | -130 | -4.6 | -260 | -4.8 |
| Shkhelda | Baksan | 13.61±0.48 | 12.50±0.50 | 8.28±0.65 | -0.31 | -1.20 | -0.72 | -950 | -36.5 | -480 | -17.1 | -1430 | -26.5 |
| Bashil | Chegem | 8.16±0.19 | 7.91±0.19 | 7.34±0.19 | -0.11 | -0.25 | -0.18 | -230 | -8.8 | -530 | -18.9 | -760 | -14.1 |
| Dolra | Enguri | 7.95±0.21 | 6.44±0.16 | 5.36±0.12 | -0.73 | -0.59 | -0.60 | -595 | -22.9 | -160 | -5.7 | -755 | -14.0 |
| | | | | | | | | | | | | **Glaciers with 1-5 km² area** | |
| Boko | Rioni | 5.07±0.12 | 4.71±0.12 | 4.62±0.11 | -0.27 | -0.07 | -0.16 | -287 | -11.0 | -436 | -15.6 | -723 | -13.4 |
| Mostotsete | Urukh Headwaters | 4.27±0.14 | 3.58±0.012 | 3.23±0.13 | -0.62 | -0.34 | -0.45 | -105 | -4.0 | -135 | 4.8 | -240 | -4.4 |
| Marukh northern | Malii Zelenchuk | 3.25±0.08 | 3.30±0.08 | 2.82±0.07 | +0.05 | -0.51 | -0.24 | -255 | -9.8 | -240 | -8.6 | -495 | -9.2 |
| Chungurjar | Ullukam | 3.13±0.09 | 2.11±0.07 | 1.88±0.07 | -1.25 | -0.39 | -0.74 | -490 | -18.8 | -405 | -14.4 | -895 | -16.6 |
| Tbilisa | Riorni | 2.90±0.10 | 2.21±0.09 | 1.91±0.08 | -0.91 | -0.48 | -0.63 | -186 | 7.2 | -354 | -12.6 | -540 | -10.0 |
| Sakeni | Kodori | 2.47±0.07 | 2.39±0.08 | 1.99±0.05 | -0.12 | -0.59 | -0.35 | -560 | -21.5 | -275 | -9.8 | -835 | -15.5 |
| Abano | Tergi | 1.96±0.09 | 1.49±0.09 | 1.33±0.09 | -0.92 | -0.38 | -0.60 | -550 | -21.2 | -240 | -8.6 | -790 | -14.6 |
| | | | | | | | | | | | | **Glaciers with <1 km² area** | |

| ID | Name | | | | | | | | | | | |
|---|---|---|---|---|---|---|---|---|---|---|---|---|
| SU5T09106388 | Rioni | 0.86±0.05 | 0.73±0.04 | 0.69±0.03 | -0.58 | -0.20 | -0.36 | -360 | -13.8 | -70 | -2.5 | -430 | -8.0 |
| *** | Sharo Argun | 0.90±0.05 | 0.77±0.04 | 0.55±0.03 | -0.55 | -1.02 | -0.72 | -65 | -2.5 | -270 | -9.6 | -335 | -6.2 |
| SU4G08011072 | Tergi | 0.62±0.03 | 0.55±0.03 | 0.42±0.02 | -0.43 | -0.84 | -0.60 | -60 | -2.3 | -310 | -11.1 | -370 | -6.9 |
| *** | Andiiskoe Koisu | 0.63±0.04 | 0.43±0.03 | 0.29±0.02 | -1.22 | -1.16 | -0.99 | -243 | -9.3 | -245 | -8.8 | -588 | -10.9 |
| SU4G08007139 | Cherek-Balkarskiy | 0.36±0.03 | 0.37±0.03 | 0.26±0.02 | +0.10 | -1.06 | -0.51 | -189 | -7.3 | -210 | -7.5 | -399 | -7.4 |
| SU4G08011083 | Tergi | 0.99±0.04 | 0.55±0.03 | 0.15±0.01 | -1.70 | -2.59 | -1.57 | -234 | -9.0 | -470 | -16.8 | -704 | -13.0 |
| SU5T09105282 | Enguri | 0.19±0.02 | 0.13±0.01 | 0.10±0.005 | -1.21 | -0.82 | -0.87 | -60 | -2.3 | -60 | -2.1 | -120 | -2.2 |

*, ** Until the 1980s the Southern and Northern Tsaneri were merged as one compound-valley type glacier. Their division likely happened in 1980-1985.

*** Omitted in WGI database.

[Figure]

**Figure 13.** Midjirgi Glacier advance between 1986-2000. (a) Landsat 5 TM, 6/08/1986. (b) Landsat 5 TM, 12/08/2000.
In 1986, the meltwater flow comes from a different position at the terminus. (c) With the snout comparison, it is visible that the snout has advanced.

[Figure]

**Figure 14.** (a) Bezingi, (b) Karaugom, (c) Kvitlodi and (d) Adishi glaciers reduction in the years 1960-1986-2014.
The 03/08/2014 Landsat 8 image (Table 1) is used as background.

---

## Author Response (AR3)

**Response letter for editor comments**

**"The Greater Caucasus Glacier Inventory (Russia/Georgia/Azerbaijan)" by L. G. Tielidze and R. D. Wheate**

http://www.the-cryosphere-discuss.net/tc-2017-48/
doi.org/10.5194/tc-2017-48

Dear Professor Chris Stokes,

Thank you for considering our manuscript "The Greater Caucasus Glacier Inventory (Russia/Georgia/Azerbaijan)". We are grateful to you and the reviewers for providing constructive feedback, which has enabled us to improve the manuscript. We have made minor changes to our manuscript following your comments we received.

All corrections are marked in yellow in the revised manuscript.

Please see answers on the comments below.

Thank you

| |
|---|
| *PAGE 1* |
| *Line 15: change to "…across the whole mountain range."* |
| We agree, please see P1 L15 |
| *Line 16: change to "…inventory at three time periods (1960, 1986, 2014) covering…"* |
| We agree, please see P1 L16 |
| *Line 18: put the date of the ASTER GDEM in brackets after it is mentioned* |
| We agree, please see P1 L18 |
| *Line 24: change to "…decreased from 0.70 km2 in 1960 to 0.66 km2 in 1986 and to 0.57 km2 in 2014".* |
| We agree, please see P1 L24-25 |
| *Line 25: change to "This new glacier inventory has been submitted to…"* |
| We agree, please see P1 L25 |
| *Line 26: delete the last sentence of the abstract: it is not a key conclusion.* |
| We agree, please see P1 L25-26 |
| *Line 38: change to "However, glacier hazards…"* |
| We agree, please see P1 L37 |
| *PAGE 2:* |
| *Line 5: delete the sentence "Our inventory has…". It seems strange to mention a key result in the Introduction* |
| We agree, please see P2 L5 |
| *PAGE 4:* |
| *Line 10: change to "1960, 1986 and 2014, including analyses of various glacier attributes* |

| |
|---|
| *(aspect, slope) and location."* |
| We agree, please see P4 L8-9 |
| *PAGE 6:*

[revised manuscript text omitted]